# Online Inverse Linear Optimization: Efficient Logarithmic-Regret Algorithm, Robustness to Suboptimality, and Lower Bound

**Shinsaku Sakaue**[*]
CyberAgent
Tokyo, Japan
shinsaku.sakaue@gmail.com

**Taira Tsuchiya**
The University of Tokyo and RIKEN AIP
Tokyo, Japan
tsuchiya@mist.i.u-tokyo.ac.jp

**Han Bao**[*]
The Institute of Statistical Mathematics
Tokyo, Japan
bao.han@ism.ac.jp

**Taihei Oki**
Hokkaido University
Hokkaido, Japan
oki@icredd.hokudai.ac.jp

## Abstract

In online inverse linear optimization, a learner observes time-varying sets of feasible actions and an agent's optimal actions, selected by solving linear optimization over the feasible actions. The learner sequentially makes predictions of the agent's true linear objective function, and their quality is measured by the *regret*, the cumulative gap between optimal objective values and those achieved by following the learner's predictions. A seminal work by Bärmann et al. (2017) obtained a regret bound of $O(\sqrt{T})$, where $T$ is the time horizon. Subsequently, the regret bound has been improved to $O(n^4 \ln T)$ by Besbes et al. (2021, 2025) and to $O(n \ln T)$ by Gollapudi et al. (2021), where $n$ is the dimension of the ambient space of objective vectors. However, these logarithmic-regret methods are highly inefficient when $T$ is large, as they need to maintain regions specified by $O(T)$ constraints, which represent possible locations of the true objective vector. In this paper, we present the first logarithmic-regret method whose per-round complexity is independent of $T$; indeed, it achieves the best-known bound of $O(n \ln T)$. Our method is strikingly simple: it applies the online Newton step (ONS) to appropriate exp-concave loss functions. Moreover, for the case where the agent's actions are possibly suboptimal, we establish a regret bound of $O(n \ln T + \sqrt{\Delta_T n \ln T})$, where $\Delta_T$ is the cumulative suboptimality of the agent's actions. This bound is achieved by using MetaGrad, which runs ONS with $\Theta(\ln T)$ different learning rates in parallel. We also present a lower bound of $\Omega(n)$, showing that the $O(n \ln T)$ bound is tight up to an $O(\ln T)$ factor.

## 1 Introduction

Optimization problems serve as forward models of various processes and systems, ranging from human decision-making to natural phenomena. In real-world applications, the true objective function of such models is rarely known a priori. This motivates the problem of inferring the objective function from observed optimal solutions, or *inverse optimization*. Early work in this area emerged from geophysics, aiming at estimating subsurface structure from seismic wave data [11, 53]. Subsequently, inverse optimization has been extensively studied [2, 13, 14, 28], applied across various domains, such as

---

[*]This work was primarily conducted during the period when SS was affiliated with the University of Tokyo and RIKEN AIP, and HB with Kyoto University and OIST.

transportation [6], power systems [9], and healthcare [12], and have laid the foundation for various machine learning methods, including inverse reinforcement learning [44] and contrastive learning [50].

This study focuses on an elementary yet fundamental case where the objective function of forward optimization is linear. We consider an *agent* who repeatedly selects an action from a set of feasible actions by solving forward linear optimization.[1] Let $n$ be a positive integer and $\mathbb{R}^n$ the ambient space where forward optimization is defined. For $t = 1, \ldots, T$, given a set $X_t \subseteq \mathbb{R}^n$ of feasible actions, the agent selects an action $x_t \in X_t$ that maximizes $x \mapsto \langle c^*, x \rangle$ over $X_t$, where $c^* \in \mathbb{R}^n$ is the agent's internal objective vector and $\langle \cdot, \cdot \rangle$ denotes the standard inner product on $\mathbb{R}^n$. We want to infer $c^*$ from observations consisting of the feasible sets and the agent's optimal actions, i.e., $\{(X_t, x_t)\}_{t=1}^{T}$.

For this problem, Bärmann et al. [4, 5] have shown that online learning methods are effective for inferring the agent's underlying objective vector $c^*$. Consider a *learner* who aims to infer $c^*$. For $t = 1, \ldots, T$, the learner makes a prediction $\hat{c}_t$ of $c^*$ based on the past observations $\{(X_i, x_i)\}_{i=1}^{t-1}$ and receives $(X_t, x_t)$ as feedback. Let $\hat{x}_t \in \arg\max_{x \in X_t} \langle \hat{c}_t, x \rangle$ represent an optimal action induced by the learner's $t$th prediction. The *regret* of choosing $\hat{x}_t$ instead of $x_t$ is defined as $\sum_{t=1}^{T} \langle c^*, x_t - \hat{x}_t \rangle$.[2] Their idea is to regard $\mathbb{R}^n \ni c \mapsto \langle c, \hat{x}_t - x_t \rangle$ as a cost function and apply online learning methods, such as the online gradient descent (OGD). Then, $\sum_{t=1}^{T} \langle c^*, x_t - \hat{x}_t \rangle \leq \sum_{t=1}^{T} \langle \hat{c}_t - c^*, \hat{x}_t - x_t \rangle = O(\sqrt{T})$ follows from the standard guarantee of online learning methods. As such, online learning methods with sublinear regret bounds can make the average regret converge to zero as $T \to \infty$.

While the regret bound of $O(\sqrt{T})$ is optimal in general online linear optimization (e.g., Hazan [26, Section 3.2]), the above online inverse linear optimization has special problem structures that could allow for better regret bounds; intuitively, feedback $(X_t, x_t)$ is more informative about $c^*$, which defines the regret, due to the optimality of $x_t \in X_t$ for $c^*$. Besbes et al. [7, 8] indeed showed that a logarithmic regret bound of $O(n^4 \ln T)$ is possible, and Gollapudi et al. [25] further improved the bound to $O(n \ln T)$.[3] While these methods significantly improve the dependence on $T$ in the regret bounds, their per-round computation cost is prohibitively high when $T$ is large. Specifically, these methods iteratively update (appropriately inflated) regions that indicate possible locations of the true objective vector $c^*$ and set prediction $\hat{c}_t$ to the "center" of the regions (the circumcenter in Besbes et al. [7, 8] and the centroid in Gollapudi et al. [25]). Since those regions are represented by $O(T)$ constraints, their per-round complexity grows polynomially in $T$, at least in a straightforward implementation. Indeed, Besbes et al. [7, 8] and Gollapudi et al. [25] only claim that their methods run in $\text{poly}(n, T)$ time. This is in stark contrast to the earlier online-learning approach of Bärmann et al. [4, 5], whose per-round complexity is independent of $T$; however, its $O(\sqrt{T})$-regret bound is much worse in terms of $T$. Is it then possible to design a logarithmic-regret method whose per-round complexity is independent of $T$?

## 1.1 Our contributions

In this paper, we first present an $O(n \ln T)$-regret method whose per-round complexity is independent of $T$ (Theorem 3.1), answering the above question affirmatively. Table 1 summarizes the comparisons of our result with prior work. Our method is very simple: we apply the online Newton step (ONS) [27] to appropriately designed exp-concave loss functions. We believe this simplicity is a strength of our method, which makes it accessible to a wider audience and easier to implement.

We then address more realistic situations where the agent's actions can be suboptimal. We establish a regret bound of $O(n \ln T + \sqrt{\Delta_T n \ln T})$, where $\Delta_T$ denotes the cumulative suboptimality of the agent's actions over $T$ rounds (Theorem 4.1). We also apply this result to the offline setting via the online-to-batch conversion (Corollary 4.2). This bound is achieved by applying MetaGrad [55, 56], a universal online learning method that runs ONS with $\Theta(\ln T)$ different learning rates in parallel, to the *suboptimality loss* [43], a loss function commonly used in inverse optimization. While universal online learning is originally intended to adapt to unknown types of loss functions, our result shows that it is useful for adapting to unknown suboptimality levels in online inverse linear optimization. At

---

[1] An "agent" is sometimes called an "expert," which we do not use to avoid confusion with the expert in universal online learning (see Section 2.3). Additionally, our results could potentially be extended to nonlinear settings based on kernel inverse optimization [6, 39], although we focus on the linear setting for simplicity.

[2] In the online setting, the learner's goal subtly deviates from inferring $c^*$ directly. Instead, the learner aims to make predictions $\hat{c}_t$ such that the induced actions $\hat{x}_t$ are good for the true objective $c^*$.

[3] Gollapudi et al. [25] studied the same problem under the name of *contextual recommendation*.

Table 1: Comparisons of the regret bound under optimal feedback and per-round/total complexity. Here, $\tau_{\text{solve}}$ is the time for computing $\hat{x}_t \in \arg\max_{x \in X_t} \langle \hat{c}_t, x \rangle$, and $\tau_{\text{E-proj}}/\tau_{\text{G-proj}}$ is the time for the Euclidean/generalized projection; typically, $\tau_{\text{E-proj}} = O(n)$ and $\tau_{\text{G-proj}} = O(n^3)$ (see Section 3 and Appendix A for details). Regarding the regret bound of Bärmann et al. [4, 5], the dependence on $n$ varies depending on the problem setting, which we discuss in Appendix A (here, set sizes are regarded as constants). Besbes et al. [7, 8] and Gollapudi et al. [25] only claim that the total complexity is $\text{poly}(n, T)$. Our inspection in Appendix A estimates the per-round complexity of Gollapudi et al. [25] as $O(\tau_{\text{solve}} + n^5 T^3)$ or higher.

|  | Regret bound | Per-round complexity | Total complexity |
|---|---|---|---|
| Bärmann et al. [4, 5] | $O(\sqrt{T})$ | $O(\tau_{\text{solve}} + \tau_{\text{E-proj}} + n)$ | Per-round $\times T$ |
| Besbes et al. [7, 8] | $O(n^4 \ln T)$ | Not claimed | $\text{poly}(n, T)$ |
| Gollapudi et al. [25] | $O(n \ln T)$ | Not claimed | $\text{poly}(n, T)$ |
| This work (Section 3) | $O(n \ln T)$ | $O(\tau_{\text{solve}} + \tau_{\text{G-proj}} + n^2)$ | Per-round $\times T$ |

a high level, our important contribution lies in uncovering the deeper connection between inverse optimization and online learning, thereby enabling the former to leverage the powerful toolkit of the latter.

Finally, we present a regret lower bound of $\Omega(n)$ (Theorem 5.1). Thus, the upper bound of $O(n \ln T)$ achieved by the method of Gollapudi et al. [25] and ours is tight up to an $O(\ln T)$ factor. While the proof idea is somewhat straightforward, this lower bound clarifies the optimal dependence on $n$ in the regret bound, thereby resolving a question raised in Besbes et al. [8, Section 7].

## 1.2 Related work

Classic studies on inverse optimization explored formulations for identifying parameters of forward optimization from a single observation [2, 29]. Recently, data-driven inverse optimization, which is intended to infer parameters of forward optimization from multiple noisy (possibly suboptimal) observations, has drawn significant interest [3, 6, 10, 35, 39, 42, 43, 52, 63]. This body of work has addressed offline settings with other criteria than the regret, which we formally define in (2). The suboptimality loss was introduced by Mohajerin Esfahani et al. [43] in this context.

The line of work by Bärmann et al. [4, 5], Besbes et al. [7, 8], and Gollapudi et al. [25], mentioned in Section 1, is the most relevant to our work; we present the detailed comparisons with them in Appendix A. It is worth mentioning here that Gollapudi et al. [25] obtained an $\exp(O(n \ln n))$-regret bound, in addition to the $O(n \ln T)$ bound in Table 1; therefore, it is possible to achieve a finite regret bound, although the dependence on $n$ is exponential. Recently, Sakaue et al. [49] obtained a finite regret bound by assuming a gap between the optimal and suboptimal objective values. Unlike their work, we do not require such gap assumptions. Online inverse linear optimization can also be viewed as a variant of stochastic linear bandits [1, 18], where noisy objective values are given as feedback, instead of optimal actions. Intuitively, the optimal-action feedback is more informative and allows for the $O(n \ln T)$ regret upper bound, while there is a lower bound of $\Omega(n\sqrt{T})$ in stochastic linear bandits [18, Theorem 3]. Online-learning approaches to other related settings have also been studied [20, 30, 59]; see Besbes et al. [8, Section 1.2] for an extensive discussion on the relation to these studies. Additionally, Chen and Kılınç-Karzan [15] and Sun et al. [51] studied online-learning methods for related settings with different criteria.

ONS [27] is a well-known online convex optimization (OCO) method that achieves a logarithmic regret bound for exp-concave loss functions. While ONS requires the prior knowledge of the exp-concavity, universal online learning methods, including MetaGrad, can automatically adapt to the unknown curvatures of loss functions, such as the strong convexity and exp-concavity [55, 56, 58, 64]. Our strategy for achieving robustness to suboptimal feedback is to combine the regret bound of MetaGrad (Proposition 2.6) with the *self-bounding* technique (see Section 4 for details), which is widely adopted in the online learning literature [23, 60, 66].

Contextual search [38, 48] is a related problem of inferring the value of $\langle c^*, x_t \rangle$ for an underlying vector $c^*$ given context vectors $x_t$. The method of Gollapudi et al. [25] is based on techniques developed in this context. Robustness to corrupted feedback is also studied in contextual search [36, 46, 47].

However, note that the problem setting is different from ours. Also, the regret in contextual search is defined with optimal choices even under corrupted feedback, and the regret bounds scale linearly with the cumulative corruption level. By contrast, our regret is defined with the agent's possibly suboptimal actions and our regret bound grows only at the rate of $\sqrt{\Delta_T}$ for the cumulative suboptimality $\Delta_T$.

Improving the per-round complexity is crucial. This topic has gathered particular attention in online portfolio selection [17, 34, 57]. There exists a trade-off between the per-round complexity and regret bounds among known methods for this problem, and advancing this frontier is recognized as important research [32, 54, 65]. When it comes to online inverse linear optimization, logarithmic regret bounds had only been achieved by the somewhat inefficient methods of Besbes et al. [7, 8] and Gollapudi et al. [25], while the efficient online-learning approach of Bärmann et al. [4, 5] only enjoys the $O(\sqrt{T})$-regret bound. This background highlights the significance of our efficient $O(n \ln T)$-regret method, which realizes the benefits of both approaches that previously existed in a trade-off relationship.

## 2 Preliminaries

### 2.1 Problem setting

We consider an online learning setting with two players, a *learner* and an *agent*. The agent sequentially solves linear optimization problems of the following form for $t = 1, \ldots, T$:

$$\text{maximize } \langle c^*, x \rangle \qquad \text{subject to } x \in X_t, \tag{1}$$

where $c^*$ is the agent's objective vector, which is unknown to the learner. Every feasible set $X_t \subseteq \mathbb{R}^n$ is non-empty and compact, and the agent's action $x_t$ always belongs to $X_t$. We assume that the agent's action is optimal for (1), i.e., $x_t \in \arg\max_{x \in X_t} \langle c^*, x \rangle$, except in Section 4, where we discuss the case where $x_t$ can be suboptimal. The set, $X_t$, is not necessarily convex; we only assume access to an oracle that returns an optimal solution $x \in \arg\max_{x' \in X_t} \langle c, x' \rangle$ for any $c \in \mathbb{R}^n$. If $X_t$ is a polyhedron, any solver for linear programs (LPs) of the form (1) can serve as the oracle. Even if (1) is, for example, an integer LP, we may use empirically efficient solvers, such as Gurobi, to obtain an optimal solution.

The learner sequentially makes a prediction of $c^*$ for $t = 1, \ldots, T$. Let $\Theta \subseteq \mathbb{R}^n$ denote a set of linear objective vectors, from which the learner picks predictions. We assume that $\Theta$ is a closed convex set and that the true objective vector $c^*$ is contained in $\Theta$. For $t = 1, \ldots, T$, the learner outputs a prediction $\hat{c}_t$ of $c^*$ based on past observations $\{(X_i, x_i)\}_{i=1}^{t-1}$ and then receives $(X_t, x_t)$ as feedback from the agent. Let $\hat{x}_t \in \arg\max_{x \in X_t} \langle \hat{c}_t, x \rangle$ denote an optimal action induced by the learner's $t$th prediction.[4] We consider the following two measures of the quality of predictions $\hat{c}_1, \ldots, \hat{c}_T \in \Theta$:

$$R_T^{c^*} := \sum_{t=1}^{T} \langle c^*, x_t - \hat{x}_t \rangle \quad \text{and} \quad \tilde{R}_T^{c^*} := R_T^{c^*} + \sum_{t=1}^{T} \langle \hat{c}_t, \hat{x}_t - x_t \rangle = \sum_{t=1}^{T} \langle \hat{c}_t - c^*, \hat{x}_t - x_t \rangle. \tag{2}$$

Following prior work [7, 8, 25], we call $R_T^{c^*}$ the *regret*, which is the cumulative gap between the optimal objective values and the objective values achieved by following the learner's predictions. Note that we have $\langle c^*, x_t - \hat{x}_t \rangle \geq 0$ as long as $x_t$ is optimal for $c^*$. While the regret is a natural performance measure, the second one, $\tilde{R}_T^{c^*}$, in (2) is convenient when considering the online-learning approach [4, 5]. We always have $R_T^{c^*} \leq \tilde{R}_T^{c^*}$ since the additional term consisting of $\langle \hat{c}_t, \hat{x}_t - x_t \rangle$ is non-negative due to the optimality of $\hat{x}_t$ for $\hat{c}_t$; intuitively, this term quantifies how well $\hat{c}_t$ explains the agent's choice $x_t$. Our upper bounds in Theorems 3.1 and 4.1 apply to $\tilde{R}_T^{c^*}$, and our lower bound in Theorem 5.1 applies to $R_T^{c^*}$.

**Remark 2.1.** The problem setting of Besbes et al. [7, 8] involves *context functions* and *initial knowledge sets*, which might make their setting appear more general than ours. However, it is not difficult to confirm that our methods are applicable to their setting. See Appendix A for details.

### 2.2 Boundedness assumptions and suboptimality loss

We introduce the following bounds on the sizes of $X_t$ and $\Theta$.

---

[4]We may break ties, if any, arbitrarily. Our results remain true as long as $\hat{x}_t$ is optimal for $\hat{c}_t$.

**Assumption 2.2.** *The $\ell_2$-diameter of $\Theta$ is bounded by $D > 0$, i.e., $\max\{\|c - c'\|_2 : c, c' \in \Theta\} \leq D$. Similarly, the $\ell_2$-diameter of $X_t$ is bounded by $K > 0$ for $t = 1, \ldots, T$. Furthermore, there exists $B > 0$ satisfying the following condition:*

$$\max\{\langle c - c', x - x'\rangle : c, c' \in \Theta, x, x' \in X_t\} \leq B \quad \text{for } t = 1, \ldots, T.$$

Assuming bounds on the diameters is common in the previous studies [4, 5, 7, 8, 25]. We additionally introduce $B > 0$ to measure the sizes of $X_t$ and $\Theta$ taking their mutual relationship into account. Note that the choice of $B = DK$ is always valid due to the Cauchy–Schwarz inequality. This quantity is inspired by a semi-norm of gradients used in Van Erven et al. [56] and enables sharper analysis than that conducted by simply setting $B = DK$.

We also define the *suboptimality loss* for later use.

**Definition 2.3.** *For $t = 1, \ldots, T$, for any action set $X_t$ and the agent's possibly suboptimal action $x_t$, the suboptimality loss is defined by $\ell_t(c) := \max_{x \in X_t}\langle c, x\rangle - \langle c, x_t\rangle$ for all $c \in \Theta$.*

That is, $\ell_t(c)$ is the suboptimality of $x_t \in X_t$ for $c$. Mohajerin Esfahani et al. [43] introduced this as a loss function that enjoys desirable computational properties in the context of inverse optimization. Specifically, the suboptimality loss is convex, and there is a convenient expression of a subgradient.

**Proposition 2.4** (cf. Bärmann et al. [4, Proposition 3.1]). *The suboptimality loss, $\ell_t : \Theta \to \mathbb{R}$, is convex. Moreover, for any $\hat{c}_t \in \Theta$ and $\hat{x}_t \in \arg\max_{x \in X_t}\langle \hat{c}_t, x\rangle$, it holds that $\hat{x}_t - x_t \in \partial\ell_t(\hat{c}_t)$.*

Confirming these properties is not difficult: the convexity is due to the fact that $\ell_t$ is the pointwise maximum of linear functions $c \mapsto \langle c, x\rangle - \langle c, x_t\rangle$, and the subgradient expression is a consequence of Danskin's theorem [19] (or one can directly prove this as in Bärmann et al. [4, Proposition 3.1]). It is worth mentioning that, as pointed out by Sakaue et al. [49], $\tilde{R}_T^{c^*}$ appears as the linearized upper bound on the regret with respect to the suboptimality loss, i.e., $\sum_{t=1}^{T}(\ell_t(\hat{c}_t) - \ell_t(c^*)) \leq \sum_{t=1}^{T}\langle \hat{c}_t - c^*, g_t\rangle = \tilde{R}_T^{c^*}$, where $g_t = \hat{x}_t - x_t \in \partial\ell_t(\hat{c}_t)$. This enables the online-to-batch conversion for the suboptimality loss, as discussed in Section 4.1. Additionally, we have $\tilde{R}_T^{c^*} = R_T^{c^*} + \sum_{t=1}^{T}\ell_t(\hat{c}_t)$ in (2).

## 2.3 ONS and MetaGrad

We briefly describe ONS and MetaGrad, based on Hazan [26, Section 4.4] and Van Erven et al. [56], to aid understanding of our methods. Appendix B shows the details for completeness. Readers who wish to proceed directly to our results may skip this subsection, taking Propositions 2.5 and 2.6 as given.

For convenience, we first state a specific form of ONS's $O(n \ln T)$ regret bound, which is later used in MetaGrad and in our analysis. See Algorithm 1 in Appendix B.1 for the pseudocode of ONS.

**Proposition 2.5.** *Let $\mathcal{W} \subseteq \mathbb{R}^n$ be a closed convex set whose $\ell_2$-diameter is at most $W > 0$. Let $w_1, \ldots, w_T$ and $g_1, \ldots, g_T$ be vectors in $\mathbb{R}^n$ satisfying the following conditions for some $G, H > 0$:*

$$w_t \in \mathcal{W}, \quad \|g_t\|_2 \leq G, \quad \text{and} \quad \max\{\langle w' - w, g_t\rangle : w, w' \in \mathcal{W}\} \leq H \quad \text{for } t = 1, \ldots, T. \quad (3)$$

*Take any $\eta \in \left(0, \frac{1}{5H}\right]$ and define loss functions $f_t^{\eta} : \mathcal{W} \to \mathbb{R}$ for $t = 1, \ldots, T$ as follows:*

$$f_t^{\eta}(w) := -\eta\langle w_t - w, g_t\rangle + \eta^2\langle w_t - w, g_t\rangle^2 \quad \text{for any } w \in \mathcal{W}. \quad (4)$$

*Let $w_1^{\eta}, \ldots, w_T^{\eta} \in \mathcal{W}$ be the outputs of ONS applied to $f_1^{\eta}, \ldots, f_T^{\eta}$. Then, for any $u \in \mathcal{W}$, it holds that*

$$\sum_{t=1}^{T}(f_t^{\eta}(w_t^{\eta}) - f_t^{\eta}(u)) = O\left(n \ln\left(\frac{WGT}{Hn}\right)\right).$$

Next, we describe MetaGrad (see Algorithm 2 in Appendix B.3), which we apply to the following general OCO problem on a closed convex set, $\mathcal{W} \subseteq \mathbb{R}^n$. For $t = 1, \ldots, T$, we select $w_t \in \mathcal{W}$ based on information obtained up to the end of round $t - 1$; then, we incur $f_t(w_t)$ and observe a subgradient, $g_t \in \partial f_t(w_t)$, where $f_t : \mathcal{W} \to \mathbb{R}$ denotes the $t$th convex loss function. We assume that $\mathcal{W}$ and $g_t$ for $t = 1, \ldots, T$ satisfy the conditions in (3). Our goal is to make the regret with respect to $f_t$, i.e., $\sum_{t=1}^{T}(f_t(w_t) - f_t(u))$, as small as possible for any comparator $u \in \mathcal{W}$.

MetaGrad maintains $\eta$-*experts*, each of whom is associated with one of $\Theta(\ln T)$ different learning rates $\eta \in \left(0, \frac{1}{5H}\right]$. Each $\eta$-expert applies ONS to loss functions $f_t^{\eta}$ of the form (4), where $w_t \in \mathcal{W}$

is the $t$th output of MetaGrad and $g_t \in \partial f_t(w_t)$ is given as feedback. In each round $t$, given the outputs $w_t^\eta$ of $\eta$-experts (which are computed based on information up to round $t-1$), MetaGrad computes $w_t \in \mathcal{W}$ by aggregating them via the exponentially weighted average (EWA).

For any comparator $u \in \mathcal{W}$, define $\tilde{R}_T^u := \sum_{t=1}^T \langle w_t - u, g_t \rangle$ and $V_T^u := \sum_{t=1}^T \langle w_t - u, g_t \rangle^2$. Since all functions $f_t$ are convex, the regret with respect to $f_t$, or $\sum_{t=1}^T (f_t(w_t) - f_t(u))$, is bounded by $\tilde{R}_T^u$ from above. Furthermore, from the definition of $f_t^\eta$, we can decompose $\tilde{R}_T^u$ as follows:

$$\tilde{R}_T^u = -\frac{\sum_{t=1}^T f_t^\eta(u)}{\eta} + \eta V_T^u = \frac{1}{\eta}\left(\sum_{t=1}^T (\overbrace{f_t^\eta(w_t)}^{\text{Zero by (4)}} - f_t^\eta(w_t^\eta)) + \sum_{t=1}^T (f_t^\eta(w_t^\eta) - f_t^\eta(u))\right) + \eta V_T^u,$$

which simultaneously holds for all $\eta > 0$. The first summation on the right-hand side, i.e., the regret of EWA compared to $w_t^\eta$, is indeed as small as $O(\ln \ln T)$, while Proposition 2.5 ensures that the second summation is $O(n \ln T)$. Thus, the right-hand side is $O\left(\frac{n \ln T}{\eta} + \eta V_T^u\right)$. If we knew the true $V_T^u$ value, we could choose $\eta \simeq \sqrt{n \ln T / V_T^u}$ to achieve $O(\sqrt{n \ln T \cdot V_T^u})$. This might seem impossible as we do not know any of $u$, $g_t$, and $w_t$ beforehand. However, we can show that at least one of $\Theta(\ln T)$ values of $\eta$ leads to almost the same regret, eschewing the need for knowing $V_T^u$. Formally, MetaGrad achieves the following regret bound (cf. Van Erven et al. [56, Corollary 8]).[5]

**Proposition 2.6.** *Let $\mathcal{W} \subseteq \mathbb{R}^n$ be given as in Proposition 2.5. Let $w_1, \dots, w_T \in \mathcal{W}$ be the outputs of MetaGrad applied to convex loss functions $f_1, \dots, f_T \colon \mathcal{W} \to \mathbb{R}$. Assume that for every $t = 1, \dots, T$, subgradient $g_t \in \partial f_t(w_t)$ satisfies the conditions (3) in Proposition 2.5. Then, it holds that*

$$\tilde{R}_T^u = O\left(\sqrt{n \ln\left(\frac{WGT}{Hn}\right) \cdot V_T^u} + Hn \ln\left(\frac{WGT}{Hn}\right)\right).$$

We outline how this result applies to exp-concave losses. Taking $W$, $G$, and $H$ to be constants and ignoring the additive term of $O(n \ln(T/n))$ for simplicity, we have $\tilde{R}_T^u = O(\sqrt{n \ln T \cdot V_T^u})$. If all $f_t$ are $\alpha$-exp-concave for some $\alpha \leq 1/(GW)$, then $f_t(w_t) - f_t(u) \leq \langle w_t - u, g_t \rangle - \frac{\alpha}{2}\langle w_t - u, g_t \rangle^2$ holds (e.g., Hazan [26, Lemma 4.3]). Summing this over $t$ and using Proposition 2.6 yield

$$\sum_{t=1}^T (f_t(w_t) - f_t(u)) \leq \tilde{R}_T^u - \frac{\alpha}{2} V_T^u = O\left(\sqrt{n \ln T \cdot V_T^u} - \alpha V_T^u\right) \lesssim O\left(\frac{n}{\alpha} \ln T\right),$$

where the last inequality is due to $\sqrt{ax} - bx \leq \frac{a}{4b}$ for any $a \geq 0$, $b > 0$, and $x \geq 0$. Remarkably, MetaGrad achieves the $O\left(\frac{n}{\alpha} \ln T\right)$ regret bound without prior knowledge of $\alpha$, whereas ONS achieves this regret bound by using the $\alpha$ value. Furthermore, even when some $f_t$ are not exp-concave, MetaGrad still enjoys a regret bound of $O(\sqrt{T} \ln \ln T)$ [56, Corollary 8]. As such, MetaGrad can automatically adapt to the unknown curvature of loss functions (at the cost of the negligible $\ln \ln T$ factor), which is the key feature of universal online learning methods.

## 3 An efficient $O(n \ln T)$-regret method based on ONS

This section presents an efficient logarithmic-regret method for online inverse linear optimization. Our method is remarkably simple: we apply ONS to exp-concave loss functions defined similarly to the $\eta$-experts' losses (4) used in MetaGrad. The proof is very short given the ONS's regret bound in Proposition 2.5. Despite this simplicity, we can achieve the regret bound of $O(n \ln T)$, which matches the best-known regret upper bound of Gollapudi et al. [25], with far lower per-round complexity.

**Theorem 3.1.** *Assume that for every $t = 1, \dots, T$, action $x_t \in X_t$ is optimal for $c^* \in \Theta$. Let $\hat{c}_1, \dots, \hat{c}_T \in \Theta$ be the outputs of ONS applied to loss functions defined as follows for $t = 1, \dots, T$:*

$$\ell_t^\eta(c) := -\eta\langle \hat{c}_t - c, \hat{x}_t - x_t \rangle + \eta^2 \langle \hat{c}_t - c, \hat{x}_t - x_t \rangle^2 \quad \text{for all } c \in \Theta, \tag{5}$$

*where $\hat{x}_t \in \arg\max_{x \in X_t} \langle \hat{c}_t, x \rangle$ and we set $\eta = \frac{1}{5B}$.[6] Then, for $R_T^{c^*}$ and $\tilde{R}_T^{c^*}$ in (2), it holds that*

$$R_T^{c^*} \leq \tilde{R}_T^{c^*} = O\left(Bn \ln\left(\frac{DKT}{Bn}\right)\right).$$

---

[5]In Van Erven et al. [56, Corollary 8], the multiplicative factor of $H$ in the second term and the denominators of $Hn$ in $\ln$ are replaced with $WG$ and $n$, respectively. We modify it to obtain the above bound; see Appendix B.

[6]This is equivalent to MetaGrad with a single $\frac{1}{5B}$-expert applied to the suboptimality losses, $\ell_1, \dots, \ell_T$.

*Proof.* Consider using Proposition 2.5 in the current setting with $\mathcal{W} = \Theta$, $w_t^\eta = w_t = \hat{c}_t$, $g_t = \hat{x}_t - x_t$, $u = c^*$, $W = D$, $G = K$, and $H = B$. Since the optimality of $x_t$ and $\hat{x}_t$ for $c^*$ and $\hat{c}_t$, respectively, ensures $\langle \hat{c}_t - c^*, \hat{x}_t - x_t \rangle \geq 0$, we have $\langle \hat{c}_t - c^*, \hat{x}_t - x_t \rangle^2 \leq B \langle \hat{c}_t - c^*, \hat{x}_t - x_t \rangle$ due to Assumption 2.2. Therefore, $\tilde{R}_T^{c^*} = \sum_{t=1}^T \langle \hat{c}_t - c^*, \hat{x}_t - x_t \rangle$ and $V_T^{c^*} := \sum_{t=1}^T \langle \hat{c}_t - c^*, \hat{x}_t - x_t \rangle^2$ satisfy $V_T^{c^*} \leq B\tilde{R}_T^{c^*}$. By using this and Proposition 2.5 with $\eta = \frac{1}{5B}$, for some constant $C_{\mathrm{ONS}} > 0$, it holds that

$$\tilde{R}_T^{c^*} = -\sum_{t=1}^T \frac{\ell_t^\eta(c^*)}{\eta} + \eta V_T^{c^*} \leq \sum_{t=1}^T \frac{\overbrace{\ell_t^\eta(\hat{c}_t)}^{\text{Zero by (5)}} - \ell_t^\eta(c^*)}{\eta} + \eta B \tilde{R}_T^{c^*} \leq 5BC_{\mathrm{ONS}} n \ln\left(\frac{DKT}{Bn}\right) + \frac{\tilde{R}_T^{c^*}}{5},$$

and rearranging the terms yields $\tilde{R}_T^{c^*} = O\big(Bn\ln\big(\frac{DKT}{Bn}\big)\big)$.[7] This also applies to $R_T^{c^*} \leq \tilde{R}_T^{c^*}$. $\qquad\square$

**Time complexity.** We discuss the time complexity of the method. Let $\tau_{\mathrm{solve}}$ be the time for solving linear optimization to find $\hat{x}_t$ and $\tau_{\mathrm{G\text{-}proj}}$ the time for the generalized projection onto $\Theta$ used in ONS (see Appendix B.1). In each round $t$, we compute $\hat{x}_t \in \arg\max_{x \in X_t} \langle \hat{c}_t, x \rangle$ in $\tau_{\mathrm{solve}}$ time; after that, the ONS update takes $O(n^2 + \tau_{\mathrm{G\text{-}proj}})$ time. Therefore, it runs in $O(\tau_{\mathrm{solve}} + n^2 + \tau_{\mathrm{G\text{-}proj}})$ time per round, which is independent of $T$. If problem (1) is an LP, $\tau_{\mathrm{solve}}$ equals the time for solving the LP (cf. Cohen et al. [16] and Jiang et al. [33]). Also, $\tau_{\mathrm{G\text{-}proj}}$ is often affordable as $\Theta$ is usually specified by the learner and hence has a simple structure. For example, if $\Theta$ is the unit Euclidean ball, the generalized projection can be computed in $O(n^3)$ time by singular value decomposition (e.g., Mhammedi et al. [41, Section 4.1]). We may also use the quasi-Newton-type method for further efficiency [40].

## 4 Robustness to suboptimal feedback with MetaGrad

In practice, assuming that the agent's actions are always optimal is unrealistic. This section discusses how to handle suboptimal feedback effectively. Here, we let $x_t \in X_t$ denote an arbitrary action taken by the agent, which the learner observes. Now that $x_t$ may have nothing to do with $c^*$, we can no longer ensure meaningful bounds on the regret that compares $\hat{x}_t$ with optimal actions. For example, if revealed actions $x_t$ remain all zeros for $t = 1, \ldots, T$, we can learn nothing about $c^*$, and hence the regret that compares $\hat{x}_t$ with optimal actions grows linearly in $T$ in the worst case. Considering this issue, we highlight that the regret, $R_T^{c^*} = \sum_{t=1}^T \langle c^*, x_t - \hat{x}_t \rangle$, used here is defined with the agent's possibly suboptimal actions $x_t$, not with those optimal for $c^*$. Small upper bounds on this regret ensure that, if the agent's actions $x_t$ are nearly optimal for $c^*$, so are $\hat{x}_t$. This regret still satisfies $R_T^{c^*} \leq \tilde{R}_T^{c^*} = \sum_{t=1}^T \langle \hat{c}_t - c^*, \hat{x}_t - x_t \rangle$ since $\hat{x}_t$ is optimal for $\hat{c}_t$. Additionally, recall that the suboptimality loss, $\ell_t$, in Definition 2.3 can be defined for any action $x_t \in X_t$ and that $\ell_t(c^*) = \max_{x \in X_t} \langle c^*, x \rangle - \langle c^*, x_t \rangle \geq 0$ indicates the suboptimality of $x_t$ for $c^*$. Below, we use $\Delta_T := \sum_{t=1}^T \ell_t(c^*)$ to denote the cumulative suboptimality of the agent's actions $x_t$.

In this setting, it is not difficult to show that ONS used in Theorem 3.1 enjoys a regret bound that scales linearly with $\Delta_T$. However, the linear dependence on $\Delta_T$ is not satisfactory, as it results in a regret bound of $O(T)$ even for small suboptimality that persists across all rounds. The following theorem ensures that by applying MetaGrad to the suboptimality losses, we can obtain a regret bound that scales with $\sqrt{\Delta_T}$.

**Theorem 4.1.** *Let $\hat{c}_1, \ldots, \hat{c}_T \in \Theta$ be the outputs of MetaGrad applied to the suboptimality losses, $\ell_1, \ldots, \ell_T$, given in Definition 2.3. Let $\hat{x}_t \in \arg\max_{x \in X_t} \langle \hat{c}_t, x \rangle$ for $t = 1, \ldots, T$. Then, it holds that*

$$R_T^{c^*} \leq \tilde{R}_T^{c^*} = O\left( Bn\ln\left(\frac{DKT}{Bn}\right) + \sqrt{\Delta_T Bn\ln\left(\frac{DKT}{Bn}\right)} \right).$$

*Proof.* Similar to the proof of Theorem 3.1, we apply Proposition 2.6 with $\mathcal{W} = \Theta$, $w_t = \hat{c}_t$, $g_t = \hat{x}_t - x_t$, $u = c^*$, $W = D$, $G = K$, and $H = B$; in addition, $g_t = \hat{x}_t - x_t \in \partial\ell_t(\hat{c}_t)$ holds due

---

[7]We may use any $\eta$ as long as $\eta B < 1$ holds; $\eta = \frac{1}{5B}$ is for consistency with MetaGrad in Appendix B.

to Proposition 2.4. Thus, Proposition 2.6 ensures the following bound for some constant $C_{\mathrm{MG}} > 0$:[8]

$$\tilde{R}_T^{c^*} \leq C_{\mathrm{MG}}\left(\sqrt{n\ln\left(\frac{DKT}{Bn}\right) \cdot V_T^{c^*}} + Bn\ln\left(\frac{DKT}{Bn}\right)\right), \tag{6}$$

where $\tilde{R}_T^{c^*} = \sum_{t=1}^T \langle \hat{c}_t - c^*, \hat{x}_t - x_t \rangle$ and $V_T^{c^*} = \sum_{t=1}^T \langle \hat{c}_t - c^*, \hat{x}_t - x_t \rangle^2$. In contrast to the case of Theorem 3.1, $\langle \hat{c}_t - c^*, \hat{x}_t - x_t \rangle^2 \leq B\langle \hat{c}_t - c^*, \hat{x}_t - x_t \rangle$ is not ensured since $\langle \hat{c}_t - c^*, \hat{x}_t - x_t \rangle$ can be negative due to the suboptimality of $x_t$. Instead, we will show that the following inequality holds:

$$\langle \hat{c}_t - c^*, \hat{x}_t - x_t \rangle^2 \leq B\langle \hat{c}_t - c^*, \hat{x}_t - x_t \rangle + 2B\ell_t(c^*). \tag{7}$$

If $\langle \hat{c}_t - c^*, \hat{x}_t - x_t \rangle \geq 0$, (7) is immediate from $\langle \hat{c}_t - c^*, \hat{x}_t - x_t \rangle^2 \leq B\langle \hat{c}_t - c^*, \hat{x}_t - x_t \rangle$ and $\ell_t(c^*) \geq 0$. If $\langle \hat{c}_t - c^*, \hat{x}_t - x_t \rangle < 0$, $\langle \hat{c}_t - c^*, \hat{x}_t - x_t \rangle^2 \leq B(-\langle \hat{c}_t - c^*, \hat{x}_t - x_t \rangle)$ holds. In addition, we have

$$\ell_t(c^*) = \max_{x \in X_t}\langle c^*, x \rangle - \langle c^*, x_t \rangle \geq \langle c^*, \hat{x}_t - x_t \rangle \geq \langle c^*, \hat{x}_t - x_t \rangle - \langle \hat{c}_t, \hat{x}_t - x_t \rangle = -\langle \hat{c}_t - c^*, \hat{x}_t - x_t \rangle,$$

where the second inequality follows from $\langle \hat{c}_t, \hat{x}_t - x_t \rangle \geq 0$. Multiplying both sides by 2 yields

$$-2\langle \hat{c}_t - c^*, \hat{x}_t - x_t \rangle \leq 2\ell_t(c^*) \iff -\langle \hat{c}_t - c^*, \hat{x}_t - x_t \rangle \leq \langle \hat{c}_t - c^*, \hat{x}_t - x_t \rangle + 2\ell_t(c^*).$$

Thus, (7) holds in any case, and hence $V_T^{c^*} \leq B\tilde{R}_T^{c^*} + 2B\Delta_T$. Substituting this into (6), we obtain

$$\tilde{R}_T^{c^*} \leq C_{\mathrm{MG}}\left(\sqrt{Bn\ln\left(\frac{DKT}{Bn}\right)\left(\tilde{R}_T^{c^*} + 2\Delta_T\right)} + Bn\ln\left(\frac{DKT}{Bn}\right)\right).$$

We assume $\tilde{R}_T^{c^*} > 0$; otherwise, the trivial bound of $\tilde{R}_T^{c^*} \leq 0$ holds. By the subadditivity of $x \mapsto \sqrt{x}$ for $x \geq 0$, we have $\tilde{R}_T^{c^*} \leq \sqrt{a\tilde{R}_T^{c^*}} + b$, where $a = C_{\mathrm{MG}}^2 Bn\ln\left(\frac{DKT}{Bn}\right)$ and $b = \sqrt{2a\Delta_T} + \frac{a}{C_{\mathrm{MG}}}$. Since $x \leq \sqrt{ax} + b$ implies $x = \frac{4}{3}x - \frac{x}{3} \leq \frac{4}{3}(\sqrt{ax} + b) - \frac{x}{3} = -\frac{1}{3}(\sqrt{x} - 2\sqrt{a})^2 + \frac{4}{3}(a + b) \leq \frac{4}{3}(a + b)$ for any $a, b, x \geq 0$, we obtain $\tilde{R}_T^{c^*} \leq \frac{4}{3}(a + b) = O\left(Bn\ln\left(\frac{DKT}{Bn}\right) + \sqrt{\Delta_T Bn\ln\left(\frac{DKT}{Bn}\right)}\right)$. $\qquad\square$

If every $x_t$ is optimal, i.e., $\Delta_T = 0$, the bound recovers that in Theorem 3.1. Note that MetaGrad requires no prior knowledge of $\Delta_T$; it automatically achieves the bound that scales with $\sqrt{\Delta_T}$, analogous to the original bound in Proposition 2.6 that scales with $\sqrt{V_T^u}$. Moreover, a refined version of MetaGrad [56] enables us to achieve a similar bound without prior knowledge of $K$, $B$, or $T$ (see Appendix B.4). Universal online learning methods shine in such scenarios where adaptivity to unknown quantities is desired. Another noteworthy point is that the last part of the proof uses the self-bounding technique [23, 60, 66]. Specifically, we derived $\tilde{R}_T^{c^*} \lesssim a + b$ from $\tilde{R}_T^{c^*} \leq \sqrt{a\tilde{R}_T^{c^*}} + b$, where the latter means that $\tilde{R}_T^{c^*}$ is upper bounded by a term of lower order in $\tilde{R}_T^{c^*}$ itself, hence the name self-bounding. We expect that the combination of universal online learning methods and self-bounding, through relations like $V_T^{c^*} \lesssim \tilde{R}_T^{c^*} + \Delta_T$ used above, will be a useful technique for deriving meaningful guarantees in online inverse linear optimization.

**Time complexity.** The use of MetaGrad comes with a slight increase in time complexity. First, as with the case of ONS, $\hat{x}_t \in \arg\max_{x \in X_t}\langle \hat{c}_t, x \rangle$ is computed in each round, taking $\tau_{\mathrm{solve}}$ time. Then, each $\eta$-expert performs the ONS update, taking $O(n^2 + \tau_{\mathrm{G\text{-}proj}})$ time. Since MetaGrad maintains $\Theta(\ln T)$ distinct $\eta$ values, the total per-round time complexity is $O\left(\tau_{\mathrm{solve}} + (n^2 + \tau_{\mathrm{G\text{-}proj}})\ln T\right)$. If the $O(\tau_{\mathrm{G\text{-}proj}}\ln T)$ factor is a bottleneck, we can use more efficient universal algorithms [41, 61] to reduce the number of projections from $\Theta(\ln T)$ to 1. Moreover, the $O(n^2)$ factor can also be reduced by sketching techniques (see Van Erven et al. [56, Section 5]).

## 4.1 Online-to-batch conversion

We briefly discuss the implication of Theorem 4.1 in the offline setting, where feedback follows some underlying distribution. As noted in Section 2.2, the bound in Theorem 4.1 applies to the regret with respect to the suboptimality loss, $\sum_{t=1}^T (\ell_t(\hat{c}_t) - \ell_t(c^*))$, since it is bounded by $\tilde{R}_T^{c^*}$ from above. Therefore, the standard online-to-batch conversion (e.g., Orabona [45, Theorem 3.1]) implies the following convergence of the average prediction in terms of the suboptimality loss.

---

[8]Here, we can simultaneously achieve $\tilde{R}_T^{c^*} = O(DK\sqrt{T\ln\ln T})$ thanks to MetaGrad's guarantee [56, Corollary 8], which can yield a stronger bound when $n$ is huge.

**Corollary 4.2.** *For any non-empty and compact $X \subseteq \mathbb{R}^n$, $x \in X$, and $c \in \Theta$, define the corresponding suboptimality loss as $\ell_{X,x}(c) := \max_{x' \in X} \langle c, x' \rangle - \langle c, x \rangle$. Let $\Delta > 0$ and define $\mathcal{X}_\Delta$ as the set of observations $(X, x)$ with bounded suboptimality, $\ell_{X,x}(c^*) \le \Delta$. Assume that $\{(X_t, x_t)\}_{t=1}^T$ are drawn i.i.d. from some distribution on $\mathcal{X}_\Delta$ (hence $\Delta_T \le \Delta T$). Let $\hat{c}_1, \ldots, \hat{c}_T \in \Theta$ be the outputs of MetaGrad applied to the suboptimality losses $\ell_t = \ell_{X_t, x_t}$ for $t = 1, \ldots, T$. Then, it holds that*

$$\mathbb{E}\left[\ell_{X,x}\left(\frac{1}{T}\sum_{t=1}^T \hat{c}_t\right) - \ell_{X,x}(c^*)\right] = O\left(\frac{Bn}{T}\ln\left(\frac{DKT}{Bn}\right) + \sqrt{\frac{\Delta Bn}{T}\ln\left(\frac{DKT}{Bn}\right)}\right).$$

Bärmann et al. [4, Theorem 3.14] also obtained a similar offline guarantee via the online-to-batch conversion. Their convergence rate is $O\left(\frac{1}{\sqrt{T}}\right)$ even when $\Delta = 0$, whereas our Corollary 4.2 offers the faster rate of $O\left(\frac{\ln T}{T}\right)$ if $\Delta = 0$. It also applies to the case of $\Delta > 0$, which is important in practice because stochastic feedback is rarely optimal at all times. We emphasize that if regret bounds scale linearly with $\Delta_T$, the above online-to-batch conversion cannot ensure that the excess suboptimality loss (the left-hand side) converge to zero as $T \to 0$. This observation lends support to the importance of the $\sqrt{\Delta_T}$-dependent regret bound we established in Theorem 4.1.

## 5 $\Omega(n)$ lower bound

We construct an instance where any online learner incurs an $\Omega(n)$ regret, implying that the $O(n \ln T)$ upper bound is tight up to an $O(\ln T)$ factor. More strongly, the following Theorem 5.1 shows that, for any $B > 0$ that gives the tight upper bound in Assumption 2.2, no learner can achieve a regret smaller than $\frac{Bn}{4}$, which means that the $Bn$ factor in our Theorem 3.1 is inevitable.

**Theorem 5.1.** *Let $n$ be a positive integer and $\Theta = \left[-\frac{1}{\sqrt{n}}, +\frac{1}{\sqrt{n}}\right]^n$. For any $T \ge n$, $B > 0$, and the learner's outputs $\hat{c}_1, \ldots, \hat{c}_T \in \Theta$, there exist $c^* \in \Theta$ and $X_1, \ldots, X_T \subseteq \mathbb{R}^n$ such that*

$$\max_{t=1,\ldots,T} \max\{\langle c - c', x - x' \rangle : c, c' \in \Theta, x, x' \in X_t\} = B \quad \text{and} \quad \mathbb{E}\left[R_T^{c^*}\right] \ge \frac{Bn}{4}$$

*hold, where $R_T^{c^*} = \sum_{t=1}^T \langle c^*, x_t - \hat{x}_t \rangle$, $x_t \in \arg\max_{x \in X_t} \langle c^*, x \rangle$, $\hat{x}_t \in \arg\max_{x \in X_t} \langle \hat{c}_t, x \rangle$, and the expectation is taken over the learner's possible randomness.*

*Proof.* We focus on the first $n$ rounds and show that any learner must incur $\frac{Bn}{4}$ in these rounds; in the remaining rounds, we may use any instance since the optimality of $x_t$ for $c^*$ ensures $\langle c^*, x_t - \hat{x}_t \rangle \ge 0$. For $t = 1, \ldots, n$, let $X_t = \left\{x \in \mathbb{R}^n : -\frac{B}{4}\sqrt{n} \le x(t) \le \frac{B}{4}\sqrt{n}, x(i) = 0 \text{ for } i \ne t\right\}$, where $x(i)$ denotes the $i$th element of $x$. That is, $X_t$ is the line segment on the $t$th axis from $-\frac{B}{4}\sqrt{n}$ to $\frac{B}{4}\sqrt{n}$. Then, $\max\{\langle c - c', x - x' \rangle : c, c' \in \Theta, x, x' \in X_t\} = B$ holds for each $t = 1, \ldots, n$. Let $c^* \in \Theta$ be a random vector such that each entry is $-\frac{1}{\sqrt{n}}$ or $\frac{1}{\sqrt{n}}$ with probability $\frac{1}{2}$, which is drawn independently of any other randomness. Then, the optimal action, $x_t \in X_t$, which is zero everywhere except that its $t$th coordinate equals $\frac{c^*(t)}{|c^*(t)|} \cdot \frac{B}{4}\sqrt{n}$, achieves $\langle c^*, x_t \rangle = \frac{B}{4}$. Note that the learner's $t$th prediction $\hat{c}_t$ is independent of $c^*(t)$ since it depends only on past observations, $\{(X_i, x_i)\}_{i=1}^{t-1}$, which have no information about $c^*(t)$. Thus, $\hat{x}_t \in \arg\max_{x \in X_t} \langle \hat{c}_t, x \rangle$ is also independent of $c^*(t)$, and hence

$$\mathbb{E}[\langle c^*, x_t - \hat{x}_t \rangle] = \mathbb{E}[\langle c^*, x_t \rangle] - \mathbb{E}[\langle c^*, \hat{x}_t \rangle] = \frac{B}{4} - \frac{1}{2}\left(-\frac{1}{\sqrt{n}} + \frac{1}{\sqrt{n}}\right)\hat{x}_t(t) = \frac{B}{4},$$

where the expectation is taken over the randomness of $c^*$. This implies that any deterministic learner incurs $\frac{Bn}{4}$ in the first $n$ rounds in expectation. Thanks to Yao's minimax principle [62], we can conclude that for any randomized learner, there exists $c^* \in \Theta$ such that $\mathbb{E}\left[R_T^{c^*}\right] \ge \frac{Bn}{4}$ holds. $\square$

In the above proof, we restricted $X_1, \ldots, X_T$ to line segments so that each $x_t \in \arg\max_{x \in X_t} \langle c^*, x \rangle$ reveals nothing about $c^*(t+1), \ldots, c^*(n)$. Whether a similar lower bound holds when all $X_t$ are full-dimensional remains an open question. Another side note is that the $\Omega(n)$ lower bound does not contradict the $O(\sqrt{T})$ upper bound of Bärmann et al. [4]. Their OGD-based method indeed achieves a regret bound of $O(DK\sqrt{T})$, where $D$ and $K$ are upper bounds on the $\ell_2$-diameters of $\Theta$ and $X_t$, respectively. In the above proof, $T \ge n$, $D \ge 1$, and $K \ge \frac{B}{2}\sqrt{n}$ hold, implying that their regret upper bound is $DK\sqrt{T} \gtrsim Bn$. Hence, the $\Omega(n)$-lower bound and their $O(DK\sqrt{T})$-upper bound are compatible.

## 6   Conclusion and discussion

We have presented an efficient ONS-based method that achieves an $O(n \ln T)$-regret bound for online inverse linear optimization. Then, we have extended the method to deal with suboptimal feedback based on MetaGrad, achieving an $O(n \ln T + \sqrt{\Delta_T n \ln T})$-regret bound, where $\Delta_T$ is the cumulative suboptimality of the agent's actions. Finally, we have presented a lower bound of $\Omega(n)$, which shows that the $O(n \ln T)$ upper bound is tight up to an $O(\ln T)$ factor. Regarding limitations, our work is restricted to the case where the agent's optimization problem is linear, as mentioned in Footnote 1; how to deal with non-linearity is an important direction for future work. In online portfolio selection, ONS is efficient but inferior to the universal portfolio algorithm regarding the dependence on the gradient norm [57]. Exploring possible similar relationships in online inverse linear optimization is left for future work. Last but not least, closing the $O(\ln T)$ gap between the upper and lower bounds is an important open problem. Interestingly, if all $X_t$ are line segments as in Section 5 and the learner can observe $X_t$ in the beginning of round $t$, the algorithm of Gollapudi et al. [25, Theorem 5.2] offers a regret upper bound of $O(n^5 \log^2 n)$, which is finite and polynomial in $n$. We also provide an additional discussion on a finite regret bound for the case of $n = 2$ in Appendix C.

### Acknowledgments and Disclosure of Funding

SS was supported by JST ERATO Grant Number JPMJER1903. TT was supported by JST ACT-X Grant Number JPMJAX210E and JSPS KAKENHI Grant Number JP24K23852. HB was supported by JST PRESTO Grant Number JPMJPR24K6. TO was supported by JST FOREST Grant Number JPMJFR232L and JSPS KAKENHI Grant Number JP22K17853.

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

# A    Detailed comparisons with previous results

Below we compare our results with Bärmann et al. [4, 5], Besbes et al. [7, 8], and Gollapudi et al. [25].

Bärmann et al. [4, 5] used $\tilde{R}_T^{c^*}$ as the performance measure, as with our Theorems 3.1 and 4.1, and provided two specific methods. The first one, based on the multiplicative weights update (MWU), is tailored for the case where $\Theta$ is the probability simplex, i.e., $\Theta = \{c \in \mathbb{R}^n \mid c \geq 0, \|c\|_1 = 1\}$. The authors assumed a bound of $K_\infty > 0$ on the $\ell_\infty$-diameters of $X_t$ and obtained a regret bound of $O(K_\infty\sqrt{T \ln n})$. The second one is based on the online gradient descent (OGD) and applies to general convex sets $\Theta$. The authors assumed that the $\ell_2$-diameters of $\Theta$ and $X_t$ are bounded by $D > 0$ and $K > 0$, respectively, and obtained a regret bound of $O(DK\sqrt{T})$. In the first case, our Theorem 3.1 with $B = K_\infty$, $D = \sqrt{2}$, and $K \leq 2\sqrt{n}K_\infty$ offers a bound of $O(K_\infty n \ln(T/\sqrt{n}))$; in the second case, we obtain a bound of $O(DKn \ln(T/n))$ by setting $B = DK$. In both cases, our bounds improve the dependence on $T$ from $\sqrt{T}$ to $\ln T$, while scaled up by a factor of $n$, up to logarithmic terms. Regarding the computation time, their MWU and OGD methods run in $O(\tau_{\text{solve}} + \tau_{\text{E-proj}} + n)$ time per round, where $\tau_{\text{E-proj}}$ is the time for the Euclidean projection onto $\Theta$, hence faster than our method. Also, suboptimal feedback is discussed in Bärmann et al. [4, Sections 3.1]. However, their bound does not achieve the logarithmic dependence on $T$ even when $\Delta_T = 0$, unlike our Theorem 4.1.

Besbes et al. [7, 8] used $R_T^{c^*}$ as the performance measure, which is upper bounded by $\tilde{R}_T^{c^*}$. They assumed that $c^*$ lies in the unit Euclidean sphere and that the $\ell_2$-diameters of $X_t$ are at most one. Under these conditions, they obtained the first logarithmic regret bound of $O(n^4 \ln T)$. By applying Theorem 3.1 to this case, we obtain a bound of $O(n \ln(T/n))$, which is better than their bound by a factor of $n^3$. As discussed in Section 1, their method relies on the idea of narrowing down regions represented with $O(T)$ constraints, and hence it seems inefficient for large $T$; indeed Besbes et al. [8, Theorem 4] only claims that the total time complexity is polynomial in $n$ and $T$. Considering this, our ONS-based method is arguably much faster while achieving the better regret bound.

**On the problem setting of Besbes et al. [7, 8].**    As mentioned in Remark 2.1, the problem setting of Besbes et al. [7, 8] is seemingly different from ours. In their setting, in each round $t$, the learner first observes $(X_t, f_t)$, where $f_t : X_t \to \mathbb{R}^n$ is called a *context function*. Then, the learner chooses $\hat{x}_t \in X_t$ and receives an optimal action $x_t \in \arg\max_{x \in X_t}\langle c^*, f_t(x)\rangle$ as feedback. It is assumed that the learner can solve $\max_{x \in X_t}\langle c, f_t(x)\rangle$ for any $c \in \mathbb{R}^n$ and that all $f_t$ are 1-Lipschitz, i.e., $\|f_t(x) - f_t(x')\|_2 \leq \|x - x'\|_2$ for all $x, x' \in X_t$. We note that our methods work in this setting, while the presence of $f_t$ might make their setting appear more general. Specifically, we redefine $X_t$ as the image of $f_t$, i.e., $\{ f_t(x) : x \in X_t \}$. Then, their assumption ensures that we can find $f_t(\hat{x}_t) \in X_t$ that maximizes $X_t \ni \xi \mapsto \langle \hat{c}_t, \xi\rangle$, and the $\ell_2$-diameter of the newly defined $X_t$ is bounded by 1 due to the 1-Lipschitzness of $f_t$. Therefore, by defining $g_t = f_t(\hat{x}_t) - f_t(x_t)$ and applying it in Theorems 3.1 and 4.1, we recover the bounds therein on $\sum_{t=1}^T \langle \hat{c}_t - c^*, f_t(\hat{x}_t) - f_t(x_t)\rangle$, with $D$, $K$, and $B$ being constants. The bounds also apply to the regret, $\sum_{t=1}^T \langle c^*, f_t(x_t) - f_t(\hat{x}_t)\rangle$, used in Besbes et al. [7, 8]. Additionally, Besbes et al. [7, 8] consider a (possibly non-convex) initial knowledge set $C_0 \subseteq \mathbb{R}^n$ that contains $c^*$. We note, however, that they do not care about whether predictions $\hat{c}_t$ lie in $C_0$ or not since the regret, their performance measure, does not explicitly involve $\hat{c}_t$. Indeed, predictions $\hat{c}_t$ that appear in their method are chosen from ellipsoidal cones that properly contain $C_0$ in general. Therefore, our methods carried out on a convex set $\Theta \supseteq C_0$ work similarly in their setting.

Gollapudi et al. [25] studied essentially the same problem as online inverse linear optimization under the name of contextual recommendation (where they and Besbes et al. [7, 8] appear to have been unaware of each other's work). As with Besbes et al. [7, 8], Gollapudi et al. [25] assumed that $c^*$ and $X_1, \ldots, X_T$ lie in the unit Euclidean ball, denoted by $\mathbb{B}^n$. Similar to Besbes et al. [7, 8], their method maintains the region $K_t$, which is the intersection of hyperplanes $\{ c \in \mathbb{R}^d : \langle c - \hat{c}_s, x_s - \hat{x}_s\rangle \geq 0 \}$ for $s = 1, \ldots, t-1$, and sets $\hat{c}_t$ to the centroid of $K_t + \frac{1}{T}\mathbb{B}^n$, where $+$ is the Minkowski sum. As regards the regret analysis, their key idea is to use the approximate Grünbaum theorem: whenever the learner incurs $\langle c^*, x_t - \hat{x}_t\rangle \geq \frac{1}{T}$, $\mathsf{Vol}\big(K_t + \frac{1}{T}\mathbb{B}^n\big)$ decreases by a constant factor, where $\mathsf{Vol}$ denotes the volume. Consequently, $\mathsf{Vol}\big(K_1 + \frac{1}{T}\mathbb{B}^n\big)/\mathsf{Vol}\big(K_T + \frac{1}{T}\mathbb{B}^n\big) \lesssim T^n$ implies the regret bound of $R_T^{c^*} = \sum_t \langle c^*, x_t - \hat{x}_t\rangle = O(n \ln T)$. As such, the per-round complexity of their method also inherently depends on $T$, and Gollapudi et al. [25, Section 1.2] only claims the total time complexity of $\mathrm{poly}(n, T)$. In this setting, our ONS-based method achieves a regret bound of $O(n \ln(T/n))$ and is arguably more efficient since the per-round complexity is independent of $T$.

---

**Algorithm 1** Online Newton Step

---

1: Set $\gamma = \frac{1}{2}\min\{\frac{1}{\beta}, \alpha\}$, $\varepsilon = \frac{n}{W^2\gamma^2}$, $A_0 = \varepsilon I_n$, and $w_1 \in \mathcal{W}$.
2: **for** $t = 1, \ldots, T$ :
3:      Play $w_t$ and observe $q_t$.
4:      $A_t \leftarrow A_{t-1} + \nabla q_t(w_t)\nabla q_t(w_t)^\top$.
5:      $w_{t+1} \leftarrow \arg\min\left\{ \left\| w_t - \frac{1}{\gamma}A_t^{-1}\nabla q_t(w_t) - w \right\|_{A_t}^2 : w \in \mathcal{W}\right\}$.    ▷ Generalized projection.

---

**Estimating the per-round complexity of Gollapudi et al. [25].** As described above, the method of Gollapudi et al. [25] requires $\hat{x}_t$ for each $t$, and hence the per-round complexity involves $\tau_{\text{solve}}$, the time to solve $\max_{x \in X_t}\langle \hat{c}_t, x\rangle$. Aside from this, its per-round complexity is dominated by the cost for computing the centroid of $K_t + \frac{1}{T}\mathbb{B}^n$, where $K_t$ is represented by $O(T)$ hyperplanes. It is known that the problem of exactly computing the centroid is #P-hard in general, but we can approximate it via sampling with a membership oracle of $K_t + \frac{1}{T}\mathbb{B}^n$. To the best of our knowledge, computing a point that is $\varepsilon$-close to the centroid takes $O(n^4/\varepsilon^2)$ membership queries, up to logarithmic factors [21, Theorem 5.7], and it is natural to set $\varepsilon = 1/T$ to make the approximation error negligible. Thus, it takes $O(n^4T^2)$ membership queries. Regarding the complexity of the membership oracle, naively checking whether a given point satisfies all the $O(T)$ linear constraints of $K_t$ takes $O(nT)$ time. Handling the Minkowski sum with $\frac{1}{T}\mathbb{B}$ would complicates the procedure, though it can be done in $\text{poly}(n, T)$ time by using, for example, Frank–Wolfe-type algorithms [22, 24, 31, 37]. For now, $O(nT)$ would be a reasonable (optimistic) estimate of the complexity of the membership oracle. Consequently, the total per-round complexity of their method is estimated to be $O(\tau_{\text{solve}} + n^5T^3)$ (or higher).

## B    Details of ONS and MetaGrad

We present the details of ONS and MetaGrad. The main purpose of this section is to provide simple descriptions and analyses of those algorithms, thereby assisting readers who are not familiar with them. As in Appendix B.4, we can also derive a regret bound of MetaGrad that yields a similar result to Theorem 4.1 directly from the results of Van Erven et al. [56].

First, we discuss the regret bound of ONS used by $\eta$-experts in MetaGrad, proving Proposition 2.5. Then, we establish the regret bound of MetaGrad in Proposition 2.6.

### B.1    Regret bound of ONS

Let $I_n \in \mathbb{R}^{n \times n}$ denote the identity matrix. For any $A, B \in \mathbb{R}^{n \times n}$, $A \succeq B$ means that $A - B$ is positive semidefinite. For positive semidefinite $A \in \mathbb{R}^{n \times n}$, let $\|x\|_A = \sqrt{x^\top A x}$ for $x \in \mathbb{R}^n$. Let $\mathcal{W} \subseteq \mathbb{R}^n$ be a closed convex set. A function $q : \mathcal{W} \to \mathbb{R}$ is $\alpha$-*exp-concave* for some $\alpha > 0$ if $\mathcal{W} \ni w \mapsto e^{-\alpha q(w)}$ is concave. For twice differentiable $q$, this is equivalent to $\nabla^2 q(w) \succeq \alpha\nabla q(w)\nabla q(w)^\top$. The following regret bound of ONS mostly comes from the standard analysis [26, Section 4.4], and hence readers familiar with it can skip the subsequent proof. The only modification lies in the use of $\beta$ instead of $W\lambda$ (defined below), where $\beta \leq W\lambda$ always holds and hence slightly tighter. This leads to the multiplicative factor of $B$, rather than $DK$, in Theorems 3.1 and 4.1.

**Proposition B.1.** *Let $\mathcal{W} \subseteq \mathbb{R}^n$ be a closed convex set with the $\ell_2$-diameter of at most $W > 0$. Assume that $q_1, \ldots, q_T : \mathcal{W} \to \mathbb{R}$ are twice differentiable and $\alpha$-exp-concave for some $\alpha > 0$. Additionally, assume that there exist $\beta, \lambda > 0$ such that $\max_{w \in \mathcal{W}}\left|\nabla q_t(w_t)^\top(w - w_t)\right| \leq \beta$ and $\|\nabla q_t(w_t)\|_2 \leq \lambda$ hold. Let $w_1, \ldots, w_T \in \mathcal{W}$ be the outputs of ONS (Algorithm 1). Then, for any $u \in \mathcal{W}$, it holds that*

$$\sum_{t=1}^{T}(q_t(w_t) - q_t(u)) \leq \frac{n}{2\gamma}\left(\ln\left(\frac{W^2\gamma^2\lambda^2 T}{n} + 1\right) + 1\right),$$

*where $\gamma = \frac{1}{2}\min\{\frac{1}{\beta}, \alpha\}$ is the parameter used in ONS.*

*Proof.* We first give a useful inequality that follows from the $\alpha$-exp-concavity. By the same analysis as the proof of Hazan [26, Lemma 4.3], for $\gamma \leq \frac{\alpha}{2}$, we have

$$q_t(w_t) - q_t(u) \leq \frac{1}{2\gamma} \ln\big(1 - 2\gamma \nabla q_t(w_t)^\top (u - w_t)\big).$$

Note that we also have $\big|2\gamma \nabla q_t(w_t)^\top (u - w_t)\big| \leq 2\gamma\beta \leq 1$. Since $\ln(1 - x) \leq -x - x^2/4$ holds for $x \geq -1$, applying this with $x = 2\gamma \nabla q_t(w_t)^\top (u - w_t)$ yields

$$q_t(w_t) - q_t(u) \leq \nabla q_t(w_t)^\top (w_t - u) - \frac{\gamma}{2}(w_t - u)^\top \nabla q_t(w_t)\nabla q_t(w_t)^\top (w_t - u). \tag{8}$$

We turn to the iterates of ONS. Since $w_{t+1}$ is the projection of $w_t - \frac{1}{\gamma}A_t^{-1}\nabla q_t(w_t)$ onto $\mathcal{W}$ with respect to the norm $\|\cdot\|_{A_t}$, we have $\|w_{t+1} - u\|_{A_t}^2 \leq \big\|w_t - \frac{1}{\gamma}A_t^{-1}\nabla q_t(w_t) - u\big\|_{A_t}^2$ for $u \in \mathcal{W}$ due to the Pythagorean theorem, hence

$$(w_{t+1} - u)^\top A_t(w_{t+1} - u)$$

$$\leq \left(w_t - \frac{1}{\gamma}A_t^{-1}\nabla q_t(w_t) - u\right)^\top A_t\left(w_t - \frac{1}{\gamma}A_t^{-1}\nabla q_t(w_t) - u\right)$$

$$= (w_t - u)^\top A_t(w_t - u) - \frac{2}{\gamma}\nabla q_t(w_t)^\top (w_t - u) + \frac{1}{\gamma^2}\nabla q_t(w_t)^\top A_t^{-1}\nabla q_t(w_t).$$

Rearranging the terms, we obtain

$$\nabla q_t(w_t)^\top (w_t - u)$$

$$\leq \frac{1}{2\gamma}\nabla q_t(w_t)^\top A_t^{-1}\nabla q_t(w_t) + \frac{\gamma}{2}(w_t - u)^\top A_t(w_t - u) - \frac{\gamma}{2}(w_{t+1} - u)^\top A_t(w_{t+1} - u).$$

From $A_t = A_{t-1} + \nabla q_t(w_t)\nabla q_t(w_t)^\top$, summing over $t$ and ignoring $\frac{\gamma}{2}(w_{T+1} - u)^\top A_T(w_{T+1} - u) \geq 0$, we obtain

$$\sum_{t=1}^T \nabla q_t(w_t)^\top (w_t - u)$$

$$\leq \frac{1}{2\gamma}\sum_{t=1}^T \nabla q_t(w_t)^\top A_t^{-1}\nabla q_t(w_t) + \frac{\gamma}{2}(w_1 - u)^\top A_1(w_1 - u)$$

$$+ \frac{\gamma}{2}\sum_{t=2}^T (w_t - u)^\top (A_t - A_{t-1})(w_t - u)$$

$$= \frac{1}{2\gamma}\sum_{t=1}^T \nabla q_t(w_t)^\top A_t^{-1}\nabla q_t(w_t) + \frac{\gamma}{2}(w_1 - u)^\top (A_1 - \nabla q_1(w_1)\nabla q_1(w_1)^\top)(w_1 - u)$$

$$+ \frac{\gamma}{2}\sum_{t=1}^T (w_t - u)^\top \nabla q_t(w_t)\nabla q_t(w_t)^\top (w_t - u).$$

Since we have $A_1 - \nabla q_1(w_1)\nabla q_1(w_1)^\top = A_0 = \varepsilon I_n$ and $\varepsilon = \frac{n}{W^2\gamma^2}$, the above inequality implies

$$\sum_{t=1}^T \nabla q_t(w_t)^\top (w_t - u) - \frac{\gamma}{2}\sum_{t=1}^T (w_t - u)^\top \nabla q_t(w_t)\nabla q_t(w_t)^\top (w_t - u)$$

$$\leq \frac{1}{2\gamma}\sum_{t=1}^T \nabla q_t(w_t)^\top A_t^{-1}\nabla q_t(w_t) + \frac{\gamma}{2}(w_1 - u)^\top A_0(w_1 - u)$$

$$\leq \frac{1}{2\gamma}\sum_{t=1}^T \nabla q_t(w_t)^\top A_t^{-1}\nabla q_t(w_t) + \frac{\gamma\varepsilon}{2}\|w_1 - u\|_2^2 \tag{9}$$

$$\leq \frac{1}{2\gamma}\sum_{t=1}^T \nabla q_t(w_t)^\top A_t^{-1}\nabla q_t(w_t) + \frac{n}{2\gamma}.$$

The first term in the right-hand side is bounded as follows due to the celebrated elliptical potential lemma (e.g., Hazan [26, proof of Theorem 4.5]):

$$\sum_{t=1}^{T} \nabla q_t(w_t)^\top A_t^{-1} \nabla q_t(w_t) \leq \ln \frac{\det A_T}{\det A_0} \leq n \ln\left(\frac{T\lambda^2}{\varepsilon} + 1\right) = n \ln\left(\frac{W^2\gamma^2\lambda^2 T}{n} + 1\right), \quad (10)$$

where we used $\det A_0 = \varepsilon^n$ and $\det A_T = \det\left(\sum_{t=1}^{T} \nabla q_t(w_t)\nabla q_t(w_t)^\top + \varepsilon I_n\right) \leq \left(T\lambda^2 + \varepsilon\right)^n$, which follows from the fact that eigenvalues of $\sum_{t=1}^{T} \nabla q_t(w_t)\nabla q_t(w_t)^\top$ are at most $T\lambda^2$. Combining (8), (9), and (10), we obtain

$$\sum_{t=1}^{T}(q_t(w_t) - q_t(u)) \leq \sum_{t=1}^{T} \nabla q_t(w_t)^\top(w_t - u) - \frac{\gamma}{2}\sum_{t=1}^{T}(w_t - u)^\top \nabla q_t(w_t)\nabla q_t(w_t)^\top(w_t - u)$$

$$\leq \frac{n}{2\gamma}\left(\ln\left(\frac{W^2\gamma^2\lambda^2 T}{n} + 1\right) + 1\right)$$

as desired. $\qquad \square$

## B.2 Regret bound of $\eta$-expert

We now establish the regret bound of ONS in Proposition 2.5, which is used by $\eta$-experts in MetaGrad. Let $\eta \in \left(0, \frac{1}{5H}\right]$ and consider applying ONS to the following loss functions, which are defined in (4):

$$f_t^\eta(w) = -\eta\langle w_t - w, g_t\rangle + \eta^2\langle w_t - w, g_t\rangle^2 \quad \text{for } t = 1, \ldots, T.$$

As in Proposition 2.5, the $\ell_2$-diameter of $\mathcal{W}$ is at most $W > 0$, and the following conditions hold:

$$w_t \in \mathcal{W}, \quad \|g_t\|_2 \leq G, \quad \text{and} \quad \max_{w,w'\in\mathcal{W}}\langle w - w', g_t\rangle \leq H \quad \text{for } t = 1, \ldots, T.$$

From $\nabla f_t^\eta(w) = \eta\left(1 - 2\eta g_t^\top(w_t - w)\right)g_t$ and $\nabla^2 f_t^\eta(w) = 2\eta^2 g_t g_t^\top$, we have

$$\nabla f_t^\eta(w)\nabla f_t^\eta(w)^\top = \eta^2\left(1 - 2\eta g_t^\top(w_t - w)\right)^2 g_t g_t^\top$$

$$\preceq \eta^2(1 + 2\eta H)^2 g_t g_t^\top = \frac{(1 + 2\eta H)^2}{2}\nabla^2 f_t^\eta(w) \quad \text{for all } w \in \mathcal{W},$$

$$\max_{w\in\mathcal{W}}\left|\nabla f_t^\eta(w_t^\eta)^\top(w - w_t^\eta)\right| = \max_{w\in\mathcal{W}}\left|\eta g_t^\top(w - w_t^\eta) - 2\eta^2\left(g_t^\top(w_t^\eta - w_t)\right)^2\right|$$

$$\leq \eta H + 2\eta^2 H^2,$$

$$\|\nabla f_t^\eta(w)\|_2 = \left\|\eta\left(1 - 2\eta g_t^\top(w_t - w)\right)g_t\right\|_2 \leq \eta(1 + 2\eta H)G.$$

Therefore, $f_t^\eta$ satisfies the conditions in Proposition B.1 with $\alpha = \frac{2}{(1+2\eta H)^2}$, $\beta = \eta H + 2\eta^2 H^2$, and $\lambda = \eta(1 + 2\eta H)G$. Since $\frac{1}{\alpha} = \frac{1}{2} + 2\eta H + 2\eta^2 H^2 \geq \beta$ holds, we have $\gamma = \frac{1}{2}\min\{\frac{1}{\beta}, \alpha\} = \frac{\alpha}{2}$. Thus, for any $\eta \in \left(0, \frac{1}{5H}\right]$, we have $\gamma \in \left[\frac{25}{49}, 1\right) \subseteq \left[\frac{1}{2}, 1\right]$ and $\gamma\lambda = \frac{\eta G}{1+2\eta H} \leq \frac{G}{7H}$. Consequently, Proposition B.1 implies that for any $u \in \mathcal{W}$, the regret of the $\eta$-expert's ONS is bounded as follows:

$$\sum_{t=1}^{T}(f_t^\eta(w_t^\eta) - f_t^\eta(u)) \leq n\left(\ln\left(\frac{W^2 G^2 T}{49nH^2} + 1\right) + 1\right) = O\left(n\ln\left(\frac{WGT}{Hn}\right)\right). \quad (11)$$

## B.3 Regret bound of MetaGrad

We turn to MetaGrad applied to convex loss functions $f_1, \ldots, f_T \colon \mathcal{W} \to \mathbb{R}$. We here use $w_t \in \mathcal{W}$ and $g_t \in \partial f_t(w_t)$ to denote the $t$th output of MetaGrad and a subgradient of $f_t$ at $w_t$, respectively, for $t = 1, \ldots, T$. We assume that these satisfy the conditions in (3), as stated in Proposition 2.6.

Algorithm 2 describes the procedure of MetaGrad. Define $\eta_i = \frac{2^{-i}}{5H}$ for $i = 0, 1, \ldots, \left\lceil\frac{1}{2}\log_2 T\right\rceil$, called *grid points*, and let $\mathcal{G} \subseteq \left(0, \frac{1}{5H}\right]$ denote the set of all grid points. For each $\eta \in \mathcal{G}$, $\eta$-expert runs ONS with loss functions $f_1^\eta, \ldots, f_T^\eta$ to compute $w_1^\eta, \ldots, w_T^\eta$. In each round $t$, we obtain $w_t$ by

---

**Algorithm 2** MetaGrad

1: $p_1^{\eta_i} \leftarrow \frac{C}{(i+1)(i+2)}$ for all $\eta_i \in \mathcal{G} = \left\{ \frac{2^{-i}}{5H} : i = 0, 1, \ldots, \left\lceil \frac{1}{2} \log_2 T \right\rceil \right\}$.
2: **for** $t = 1, \ldots, T$ :
3:      Fetch $w_t^\eta$ from $\eta$-experts for all $\eta \in \mathcal{G}$.
4:      Play $w_t = \frac{\sum_{\eta \in \mathcal{G}} \eta p_t^\eta w_t^\eta}{\sum_{\eta \in \mathcal{G}} \eta p_t^\eta}$.
5:      Observe $g_t \in \partial f_t(w_t)$ and send $(w_t, g_t)$ to $\eta$-experts for all $\eta \in \mathcal{G}$.
6:      $p_{t+1}^\eta \leftarrow p_t^\eta \exp(-f_t^\eta(w_t^\eta))/Z_t$ for all $\eta \in \mathcal{G}$, where $Z_t = \sum_{\eta \in \mathcal{G}} p_t^\eta \exp(-f_t^\eta(w_t^\eta))$.

---

aggregating the $\eta$-experts' outputs $w_t^\eta$ based on the exponentially weighted average method (EWA). We set the prior as $p_1^{\eta_i} = \frac{C}{(i+1)(i+2)}$ for all $\eta_i \in \mathcal{G}$, where $C = 1 + \frac{1}{1 + \left\lceil \frac{1}{2} \log_2 T \right\rceil}$. Then, it is known that for every $\eta \in \mathcal{G}$, the regret of EWA relative to the $\eta$-expert's choice $w_t^\eta$ is bounded as follows:

$$\sum_{t=1}^{T} (f_t^\eta(w_t) - f_t^\eta(w_t^\eta)) \leq \ln \frac{1}{p_1^\eta} \leq \ln \left( \left( \left\lceil \frac{1}{2} \log_2 T \right\rceil + 1 \right) \left( \left\lceil \frac{1}{2} \log_2 T \right\rceil + 2 \right) \right)$$
$$\leq 2 \ln \left( \frac{1}{2} \log_2 T + 3 \right),$$

(12)

where we used $C \geq 1$ in the second inequality. We here omit the proof as it is completely the same as that of Van Erven and Koolen [55, Lemma 4] (see also Wang et al. [58, Lemma 1]).

We are ready to prove Proposition 2.6. Let $V_T^u = \sum_{t=1}^{T} \langle w_t - u, g_t \rangle^2$. By using $f_t^\eta(w_t) = 0$, (11), and (12), it holds that

$$\sum_{t=1}^{T} \langle w_t - u, g_t \rangle = -\frac{\sum_{t=1}^{T} f_t^\eta(u)}{\eta} + \eta V_T^u$$

$$= \frac{1}{\eta} \left( \underbrace{\sum_{t=1}^{T} (f_t^\eta(w_t) - f_t^\eta(w_t^\eta))}_{\text{Regret of EWA w.r.t. } w_t^\eta} + \underbrace{\sum_{t=1}^{T} (f_t^\eta(w_t^\eta) - f_t^\eta(u))}_{\text{Regret of } \eta\text{-expert w.r.t. } u} \right) + \eta V_T^u$$

$$\leq \frac{1}{\eta} \left( 2 \ln \left( \frac{1}{2} \log_2 T + 3 \right) + n \left( \ln \left( \frac{W^2 G^2 T}{49 n H^2} + 1 \right) + 1 \right) \right) + \eta V_T^u$$

for all $\eta \in \mathcal{G}$. For brevity, let

$$A = 2 \ln \left( \frac{1}{2} \log_2 T + 3 \right) + n \left( \ln \left( \frac{W^2 G^2 T}{49 n H^2} + 1 \right) + 1 \right) \geq 1.$$

If we knew $V_T^u$, we could set $\eta$ to $\eta^* := \sqrt{\frac{A}{V_T^u}} \geq \frac{1}{5H\sqrt{T}}$ to minimize the above regret bound, $\frac{A}{\eta} + \eta V_T^u$. Actually, we can do almost the same without knowing $V_T^u$ thanks to the fact that the regret bound holds for all $\eta \in \mathcal{G}$. If $\eta^* \leq \frac{1}{5H}$, by construction we have a grid point $\eta \in \mathcal{G}$ such that $\eta^* \in \left[ \frac{\eta}{2}, \eta \right]$, hence

$$\sum_{t=1}^{T} \langle w_t - u, g_t \rangle \leq \eta V_T^u + \frac{A}{\eta} \leq 2\eta^* V_T^u + \frac{A}{\eta^*} \leq 3 \sqrt{A V_T^u}.$$

Otherwise, $\eta^* = \sqrt{\frac{A}{V_T^u}} \geq \frac{1}{5H}$ holds, which implies $V_T^u \leq 25 H^2 A$. Thus, for $\eta_0 = \frac{1}{5H} \in \mathcal{G}$, we have

$$\sum_{t=1}^{T} \langle w_t - u, g_t \rangle \leq \eta_0 V_T^u + \frac{A}{\eta_0} \leq 10 H A.$$

Therefore, in any case, we have

$$\sum_{t=1}^{T} \langle w_t - u, g_t \rangle \leq 3 \sqrt{A V_T^u} + 10 H A = O\left( \sqrt{n \ln \left( \frac{WGT}{Hn} \right) \cdot V_T^u} + H n \ln \left( \frac{WGT}{Hn} \right) \right),$$

obtaining the regret bound in Proposition 2.6.

**Algorithm 3** $O(1)$-Regret Algorithm for $n = 2$.

1: Set $\mathcal{C}_1$ to $\mathbb{S}^1$.
2: **for** $t = 1, \ldots, T$ **:**
3:     Draw $\hat{c}_t$ uniformly at random from $\mathcal{C}_t$.
4:     Observe $(X_t, x_t)$.
5:     $\mathcal{C}_{t+1} \leftarrow \mathcal{C}_t \cap \mathcal{N}_t$. ▷ $\mathcal{N}_t$ is the normal cone.

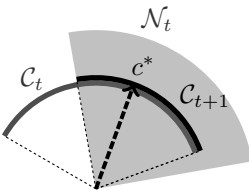

Figure 1: Illustration of $c^*$, $\mathcal{C}_t$, $\mathcal{N}_t$, and $\mathcal{C}_{t+1}$.

### B.4 Lipschitz adaptivity and anytime guarantee

Recent studies [41, 56] have shown that MetaGrad can be further made Lipschitz adaptive and agnostic to the number of rounds. Specifically, MetaGrad given in Van Erven et al. [56, Algorithms 1 and 2] works without knowing $G$, $H$, or $T$ in advance, while using (a guess of) $W$. By expanding the proofs of Van Erven et al. [56, Theorem 7 and Corollary 8], we can confirm that the refined version of MetaGrad enjoys the following regret bound:

$$\sum_{t=1}^{T} \langle w_t - u, g_t \rangle = O\left( \sqrt{n \ln\left( \frac{WGT}{n} \right) \cdot V_T^u} + Hn \ln\left( \frac{WGT}{n} \right) \right).$$

By using this in the proof of Theorem 4.1, we obtain

$$\sum_{t=1}^{T} \langle c^*, x_t - \hat{x}_t \rangle \leq \sum_{t=1}^{T} \langle \hat{c}_t - c^*, \hat{x}_t - x_t \rangle = O\left( Bn \ln\left( \frac{DKT}{n} \right) + \sqrt{\Delta_T Bn \ln\left( \frac{DKT}{n} \right)} \right),$$

and the algorithm does not require knowing $K$, $B$, $T$, or $\Delta_T$ in advance.

## C    On removing the $\ln T$ factor: the case of $n = 2$

This section provides an additional discussion on closing the $\ln T$ gap in the upper and lower bounds on the regret. Specifically, focusing on the case of $n = 2$, we provide a simple algorithm that achieves a regret bound of $O(1)$ in expectation, removing the $\ln T$ factor. We also observe that extending the algorithm to general $n \geq 2$ might be challenging. Note that Gollapudi et al. [25, Theorem 4.1] has already established a regret bound of $\exp(O(n \ln n))$ as mentioned in Section 1.2, which implies an $O(1)$-regret bound for $n = 2$. The purpose of this section is simply to stimulate discussions on closing the $\ln T$ gap by presenting another simple analysis. Below, let $\mathbb{B}^n$ and $\mathbb{S}^{n-1}$ denote the unit Euclidean ball and sphere in $\mathbb{R}^n$, respectively, for any integer $n > 1$.

### C.1    An $O(1)$-regret method for $n = 2$

We focus on the case of $n = 2$ and present an algorithm that achieves a regret bound of $O(1)$ in expectation. We assume that all $x_t \in X_t$ are optimal for $c^*$ for $t = 1, \ldots, T$. For simplicity, we additionally assume that all $X_t$ are contained in $\frac{1}{2}\mathbb{B}^2$ and that $c^*$ lies in $\mathbb{S}^1$. For any non-zero vectors $c, c' \in \mathbb{R}^n$, let $\theta(c, c')$ denote the angle between the two vectors. The following lemma from Besbes et al. [8], which holds for general $n \geq 2$, is useful in the subsequent analysis.

**Lemma C.1** (Besbes et al. [8, Lemma 1]). *Let $c^*, \hat{c}_t \in \mathbb{S}^{n-1}$, $X_t \subseteq \frac{1}{2}\mathbb{B}^n$, $x_t \in \arg\max_{x \in X_t} \langle c^*, x \rangle$, and $\hat{x}_t \in \arg\max_{x \in X_t} \langle \hat{c}_t, x \rangle$. If $\theta(c^*, \hat{c}_t) < \pi/2$, it holds that $\langle c^*, x_t - \hat{x}_t \rangle \leq \sin \theta(c^*, \hat{c}_t)$.*

Our algorithm, given in Algorithm 3, is a randomized variant of the one investigated by Besbes et al. [7, 8]. The procedure is intuitive: we maintain a set $\mathcal{C}_t \subseteq \mathbb{S}^1$ that contains $c^*$, from which we draw $\hat{c}_t$ uniformly at random, and update $\mathcal{C}_t$ by excluding the area that is ensured not to contain $c^*$ based on the $t$th feedback $(X_t, x_t)$. Formally, the last step takes the intersection of $\mathcal{C}_t$ and the *normal cone* $\mathcal{N}_t = \{ c \in \mathbb{R}^n : \langle c, x_t - x \rangle \geq 0, \forall x \in X_t \}$ of $X_t$ at $x_t$, which is a convex cone containing $c^*$. Therefore, every $\mathcal{C}_t$ is a connected arc on $\mathbb{S}^1$ and is non-empty due to $c^* \in \mathcal{C}_t$ (see Figure 1).

**Theorem C.2.** *For the above setting of $n = 2$, Algorithm 3 achieves $\mathbb{E}[R_T^{c^*}] \leq 2\pi$.*

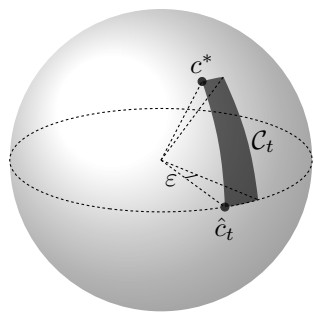

Figure 2: An example of $\mathcal{C}_t$ on $\mathbb{S}^2$. The darker area, $A(\mathcal{C}_t)$, becomes arbitrarily small as $\varepsilon \to 0$, while $\theta(c^*, \hat{c}_t)$ does not.

*Proof.* For any connected arc $\mathcal{C} \subseteq \mathbb{S}^1$, let $A(\mathcal{C}) \in [0, 2\pi]$ denote its central angle, which equals its length. Fix $\mathcal{C}_t$. If $\hat{c}_t \in \mathcal{C}_t \cap \mathrm{int}(\mathcal{N}_t)$, where $\mathrm{int}(\cdot)$ denotes the interior, $\hat{x}_t = x_t$ is the unique optimal solution for $\hat{c}_t$, hence $\langle c^*, x_t - \hat{x}_t \rangle = 0$. Taking the expectation about the randomness of $\hat{c}_t$, we have

$$\mathbb{E}[\langle c^*, x_t - \hat{x}_t \rangle] = \Pr[\hat{c}_t \in \mathcal{C}_t \setminus \mathrm{int}(\mathcal{N}_t)] \, \mathbb{E}[\langle c^*, x_t - \hat{x}_t \rangle \mid \hat{c}_t \in \mathcal{C}_t \setminus \mathrm{int}(\mathcal{N}_t)]$$

$$= \frac{A(\mathcal{C}_t \setminus \mathcal{N}_t)}{A(\mathcal{C}_t)} \, \mathbb{E}[\langle c^*, x_t - \hat{x}_t \rangle \mid \hat{c}_t \in \mathcal{C}_t \setminus \mathrm{int}(\mathcal{N}_t)],$$

where we used $\Pr[\hat{c}_t \in \mathcal{C}_t \setminus \mathrm{int}(\mathcal{N}_t)] = \Pr[\hat{c}_t \in \mathcal{C}_t \setminus \mathcal{N}_t] = A(\mathcal{C}_t \setminus \mathcal{N}_t)/A(\mathcal{C}_t)$ (since the boundary of $\mathcal{N}_t$ has zero measure). If $A(\mathcal{C}_t) \geq \pi/2$, from $\langle c^*, x_t - \hat{x}_t \rangle \leq \|c^*\|_2 \|x_t - \hat{x}_t\|_2 \leq 1$, we have

$$\mathbb{E}[\langle c^*, x_t - \hat{x}_t \rangle] \leq \frac{2}{\pi} A(\mathcal{C}_t \setminus \mathcal{N}_t) \leq A(\mathcal{C}_t \setminus \mathcal{N}_t).$$

If $A(\mathcal{C}_t) < \pi/2$, Lemma C.1 and $\hat{c}_t, c^* \in \mathcal{C}_t$ imply $\langle c^*, x_t - \hat{x}_t \rangle \leq \sin \theta(c^*, \hat{c}_t) \leq \sin A(\mathcal{C}_t)$. Thus, by using $\frac{1}{x} \sin x \leq 1$ ($x \in \mathbb{R}$), we obtain

$$\mathbb{E}[\langle c^*, x_t - \hat{x}_t \rangle] \leq \frac{A(\mathcal{C}_t \setminus \mathcal{N}_t)}{A(\mathcal{C}_t)} \sin A(\mathcal{C}_t) \leq A(\mathcal{C}_t \setminus \mathcal{N}_t).$$

Therefore, we have $\mathbb{E}[\langle c^*, x_t - \hat{x}_t \rangle] \leq A(\mathcal{C}_t \setminus \mathcal{N}_t)$ in any case. Consequently, we obtain

$$\mathbb{E}\left[R_T^{c^*}\right] = \sum_{t=1}^{T} \mathbb{E}[\langle c^*, x_t - \hat{x}_t \rangle] \leq \sum_{t=1}^{T} A(\mathcal{C}_t \setminus \mathcal{N}_t) \leq 2\pi,$$

where the last inequality is due to $\mathcal{C}_{t+1} = \mathcal{C}_t \cap \mathcal{N}_t$, which implies $\mathcal{C}_s \subseteq \mathcal{C}_t$ and $\mathcal{C}_s \cap (\mathcal{C}_t \setminus \mathcal{N}_t) = \emptyset$ for any $s > t$, and hence no double counting occurs in the above summation. $\square$

## C.2 Discussion on higher-dimensional cases

Algorithm 3 might appear applicable to general $n \geq 2$ by replacing $\mathbb{S}^1$ with $\mathbb{S}^{n-1}$ and defining $A(\mathcal{C}_t)$ as the area of $\mathcal{C}_t \subseteq \mathbb{S}^{n-1}$. However, this idea faces a challenge in bounding the regret when extending the above proof to general $n \geq 2$.[9]

As suggested in the proof of Theorem C.2, bounding $\mathbb{E}[\langle c^*, x_t - \hat{x}_t \rangle]$ is trickier when $A(\mathcal{C}_t)$ is small (cf. the case of $A(\mathcal{C}_t) < \pi/2$). Luckily, when $n = 2$, we can bound it thanks to Lemma C.1 and $\sin \theta(c^*, \hat{c}_t) \leq \sin A(\mathcal{C}_t)$, where the latter roughly means the angle, $\theta(c^*, \hat{c}_t)$, is bounded by the area, $A(\mathcal{C}_t)$, from above. Importantly, when $n = 2$, both the central angle and the area of an arc are identified with the length of the arc, which is the key to establishing $\sin \theta(c^*, \hat{c}_t) \leq \sin A(\mathcal{C}_t)$. This is no longer true for $n \geq 3$. As in Figure 2, the area, $A(\mathcal{C}_t)$, can be arbitrarily small even if the angle

---

[9]We note that a hardness result given in Besbes et al. [8, Theorem 2] is different from what we encounter here. They showed that their *greedy circumcenter policy* fails to achieve a sublinear regret, which stems from the shape of the initial knowledge set and the behavior of the greedy rule for selecting $\hat{c}_t$; this differs from the issue discussed above.

within there, or the maximum $\theta(c^*, \hat{c}_t)$ for $c^*, \hat{c}_t \in \mathcal{C}_t$, is large.[10] This is why the proof for the case of $n = 2$ does not directly extend to higher dimensions. We leave closing the $O(\ln T)$ gap for $n \geq 3$ as an important open problem for future research.

## D  Numerical experiments

We conducted numerical experiments to complement our theoretical results. Experiments were conducted on Google Colab equipped with an Intel® Xeon® CPU @ 2.20GHz, 12 GB RAM, running Ubuntu 22.04.4 LTS with Python 3.12.11. The code is available at https://github.com/ssakaue/online-inverse-linear-optimization-code.

We use a setup based on the hard instance considered in our lower bound analysis (Section 5). Let $c^*$ be a random vector with $\|c^*\|_2 = 1$. The learner's prediction set $\Theta$ is the $n$-dimensional Euclidean unit ball. At each round $t$, we sample an endpoint $v$ uniformly at random from the unit sphere in $\mathbb{R}^n$, and set $X_t = \{-v, +v\}$. We report results for $T = 10{,}000$ rounds in dimensions $n = 2$, 20, and 200. To mitigate randomness, we repeat each experiment 10 times independently and report mean and standard deviation.

**Compared methods.**  We compare the following three methods:

- **ONS**: our proposed method based on ONS,
- **OGD**: the OGD-based method of Bärmann et al. [4],
- **CP**: a cutting-plane style method inspired by Gollapudi et al. [25].

For CP, computing the centroid of the feasible region is #P-hard. Therefore, we adopt a randomized heuristic: we pre-sample $n \times 10^4$ candidate points from the unit ball, eliminate those violating accumulated cuts, and approximate the centroid by averaging the remaining points.

**Results.**  Table 2 summarizes the cumulative regret and runtime over $T = 10{,}000$ rounds (mean $\pm$ standard deviation across 10 trials).

Table 2: Experimental results over $T = 10{,}000$ rounds (mean $\pm$ standard deviation across 10 independent runs).

(a) Cumulative Regret

| Method | $n = 2$ | $n = 20$ | $n = 200$ |
|---|---|---|---|
| ONS | $7.81 \pm 2.52$ | $31.55 \pm 0.43$ | $46.19 \pm 0.88$ |
| OGD | $8.46 \pm 4.07$ | $33.80 \pm 0.50$ | $67.41 \pm 1.36$ |
| CP | $2.77 \pm 0.68$ | $829.91 \pm 287.00$ | $515.50 \pm 43.75$ |

(b) Cumulative Runtime (s)

| Method | $n = 2$ | $n = 20$ | $n = 200$ |
|---|---|---|---|
| ONS | $0.648 \pm 0.115$ | $0.705 \pm 0.086$ | $9.073 \pm 0.727$ |
| OGD | $0.150 \pm 0.024$ | $0.150 \pm 0.017$ | $0.233 \pm 0.019$ |
| CP | $4.670 \pm 3.205$ | $5.700 \pm 1.748$ | $71.252 \pm 12.152$ |

When the dimension is small ($n = 2$), CP achieves the lowest cumulative regret. However, its cumulative regret deteriorates significantly for $n = 20$ and $n = 200$. This degradation stems from the limited number of pre-sampled candidate points: in principle, about $T^n$ samples would be required for an accurate approximation, which is infeasible even for moderate $n$. Moreover, although the above randomized centroid approximation substantially reduces computation by averaging over the

---

[10]A similar issue, though leading to different challenges, is noted in Besbes et al. [8, Section 4.4], where their method encounters ill-conditioned (or elongated) ellipsoids. They addressed this by appropriately determining when to update the ellipsoidal cone. The $\ln T$ factor arises as a result of balancing being ill-conditioned with the instantaneous regret.

surviving candidates, it remains less scalable than OGD and ONS. These results highlight practical limitations of CP in moderate to high-dimensional settings.

In contrast, ONS consistently achieves low regret values across all dimensions while remaining computationally feasible. It outperforms OGD in terms of the regret and scales reasonably well with increasing $n$. Note that our ONS implementation uses a straightforward projection subroutine that repeatedly solves similar linear systems—an overhead that could be reduced by more sophisticated implementation techniques. Further speedups could also be achieved via quasi-Newton-type updates or sketching-based techniques, as discussed in Sections 3 and 4.

Taken together, these findings affirm that ONS provides a strong and scalable alternative to existing methods in online inverse linear optimization, especially when CP is computationally infeasible and OGD's regret performance is unsatisfactory.

