# OpenReview forum: "Online Inverse Linear Optimization: Efficient Logarithmic-Regret Algorithm, Robustness to Suboptimality, and Lower Bound"
_NeurIPS.cc/2025/Conference — NeurIPS 2025 poster_

### Official Review · Reviewer_Td8g · 2025-06-25

**Clarity:** 4
**Significance:** 3
**Originality:** 2
**Rating:** 5
**Confidence:** 3

**Summary:**

This paper addresses the problem of online inverse linear optimization. In this framework, at each round, the learner receives a feasible optimization set along with the optimal result of an oracle call (referred to as the agent). The learner aims to recover the underlying linear parameter from these observations by minimizing cumulative regret.

Previous work has shown that a black-box reduction to online convex optimization can be applied, yielding regret upper bounds of order $O(\sqrt{T})$ or $O(n \log T)$. However, the logarithmic regret rate was only achieved by methods with a per-round time complexity of $\Omega(T^3)$, which is polynomial and thus impractical for large $T$.

In this work, the authors present a simple reduction that allows the use of Online Newton Step (ONS) with a loss function inspired by that of MetaGrad. This approach achieves a regret of order $O(n \log T)$ while retaining the per-round computational complexity of ONS---i.e., constant per round, with the main computational bottleneck being the generalized projection step.

Moreover, by applying MetaGrad instead of ONS---which automatically tunes the parameters of ONS---they show that their method remains robust even when the agent is suboptimal. Specifically, their regret scales as $O(\sqrt{\Delta_T})$ rather than linearly, where $\Delta_T$ denotes the cumulative suboptimality of the agent.

Finally, the authors provide an $\Omega(n)$ lower bound (on a specific instance), demonstrating the optimality of their method up to a logarithmic factor in $T$.

**Questions:**

1) For the online-to-batch conversion, is it possible to obtain the result with high probability?

2) Since MetaGrad simultaneously achieves both $DK\sqrt{T}$ and $n \log T$ regret bounds, it would be valuable to state this explicitly in your results. The $DK\sqrt{T}$ bound may be preferable in regimes where $n$ is large.

**Ethical Concerns:**

["NO or VERY MINOR ethics concerns only"]

**Final Justification:**

I went through the reviews and rebuttal and support acceptance. In my view, the central question is whether the limited novelty of the analysis (being a straightforward application of ONS and MetaGrad) should be seen as a strength or a weakness. I believe it is a strength, as it improves previous approaches, that were significantly more complex. I think the authors address well other concerns in their rebuttal.

**Limitations:**

Yes

**Paper Formatting Concerns:**

NC

**Quality:**

3

**Strengths And Weaknesses:**

Strengths:

The paper is well-written, and the analysis is, as the authors claim, relatively simple. It mainly involves defining an appropriate loss function and applying existing results from ONS and MetaGrad. This simplicity can be viewed as a strength (a straightforward method with easy implementation that significantly improves upon existing results) or as a weakness (the lack of novel technical contributions).


Weaknesses:

The paper lacks numerical experiments. It would have been valuable to demonstrate, in practice, the benefits in terms of regret or computational cost using synthetic experiments. In particular, what is the effective cost of the generalized projection step in ONS when solving the optimization problem to sufficient accuracy for the regret guarantees? (It would be helpful to recall this in the paper.) If both algorithms are given similar time budgets (e.g., by controlling the precision with which the optimization problems are solved), would it be possible to compare with the regret bound of [25]?

There remains a gap between the upper and lower bounds: $\Omega(n)$ versus $O(n \log T)$. Moreover, constant regret might be achievable in general, although it is likely to be challenging. It would also be worthwhile to include in the comparison table the constant regret bound (albeit exponential in $n$) from [25]. Is it possible that an exponential dependence on $n$ is unavoidable for achieving constant regret?

---

> ### Author Rebuttal · Authors · 2025-07-25
>
> We sincerely thank the reviewer for the positive feedback and insightful questions. Below, we address each point raised.
>
> ## **W1. Numerical experiments**
> Thank you for the constructive suggestion regarding numerical experiments. Encouraged by your and other reviewers' comments, we conducted preliminary experiments to assess the practical performance of our approach.
>
> In short, we found that cutting-plane-style methods (CP; e.g., Besbes et al. 2021, 2025; Gollapudi et al. 2021) quickly become computationally impractical, even in moderate dimensions. In contrast, our ONS-based method remains tractable and consistently outperforms OGD (Bärmann et al., 2020) in terms of the regret. The results provide empirical evidence that ONS can serve as a strong practical alternative to both CP and OGD. Please see our response to Reviewer xL4W for the details of the experimental setup and results.
>
> Regarding the effective cost of the generalized projection in ONS, as discussed by Mhammedi et al. (2019), the projection can be implemented via singular value decomposition (SVD), which achieves machine epsilon precision in $O(n^3)$ time. On the other hand, cutting-plane-based methods such as that of Gollapudi et al. [25] require an impractically large number of samples to reach comparable approximation accuracy. Thus, under any realistic time budget, ONS would perform significantly better than CP in terms of the regret—an observation supported by our experiments for $n = 20$ and $n = 200$.
>
> ## **W2. On the dependence on $n$ in the constant regret bound**
> Whether it is possible to avoid the exponential dependence on $n$ in the constant regret bound remains an important open question. This is highlighted in Section 6 of our paper and in Gollapudi et al. [25, Section 4.1]. While prior approaches that achieve logarithmic and $\exp(O(n \log n))$ regret bounds (Gollapudi et al. 2021; Besbes et al. 2021, 2025) rely on cutting-plane methods, our work introduces a new approach based on Online Newton Step (ONS). The insights from these distinct methodologies would be complementary, and we hope that combining them will lead to progress on this open problem. In this respect, we believe that our ONS-based approach provides a meaningful contribution.
>
>
> ## **Q1. High-probability online-to-batch conversion**
> Yes, since it is the application of the standard online-to-batch conversion to the convex suboptimality loss, it can be extended to a high-probability guarantee via a commonly used concentration argument (e.g., Orabona "A Modern Introduction to Online Learning," Section 3.1.2). A caveat is that, as is usual, obtaining a guarantee that holds with probability at least $1 - \delta$ comes with an additive $O(\sqrt{\log(1/\delta)/T})$ term. This additive factor makes the overall convergence rate $O(\log T / T + \sqrt{\log(1/\delta)/T})$, which is a fundamental limitation of the online-to-batch conversion approach to obtaining high-probability guarantees.
>
> ## **Q2. Making both regret bounds explicit**
> We appreciate the suggestion. We agree that explicitly stating the $O(DK\sqrt{T})$ regret bound alongside the logarithmic one would be helpful. We will incorporate it into the revised version.
>
> We hope that the above responses effectively address the points raised. Please do not hesitate to let us know if any questions remain or if further clarification would be helpful.

---

### Official Review · Reviewer_xL4W · 2025-07-01

**Clarity:** 3
**Significance:** 2
**Originality:** 3
**Rating:** 4
**Confidence:** 4

**Summary:**

This paper studies the problem of online inverse linear optimization. By applying the Online Newton Step (ONS) algorithm, the authors achieve logarithmic regret in time ($\log T$) with time independent per round complexity. The setting is further extended to account for suboptimal actions, and corresponding lower bounds are established.

**Questions:**

Is the suboptimal feedback setting considered in the paper related in any way to the framework of online optimization with hints? If so, it would be helpful to clarify the connection and highlight any similarities or differences.

**Ethical Concerns:**

["NO or VERY MINOR ethics concerns only"]

**Final Justification:**

The authors adequately addressed my questions and I increased the initial score.

**Limitations:**

Adequately discussed.

**Quality:**

3

**Strengths And Weaknesses:**

Strengths:

- Per round complexity of different algorithms are compared and time independent per round complexity has been achieved.

- Lower bound of regret in terms of dimensional dependence has been proved.

Weaknesses:

- It seems that there is a tradeoff between per round complexity and regret bound in Table 1. In the case of high dimensions ONS can be costly which can also be observed in $n^2$ term in per round complexity.

- The reported overall complexity can be somewhat misleading. For example, $\tau_{proj}$ itself can depend on $n$, in which case the overall complexity may be well more expensive that just $n^2$.

- While the paper motivates the results from a practical perspective, there is no numerical experiment to demonstrate the effectiveness of the proposed methods over state-of-the-art.

---

> ### Author Rebuttal · Authors · 2025-07-25
>
> We sincerely thank the reviewer for their time and effort in reviewing our work. Below, we address each point raised.
>
> ## **W1 & W2. Regarding complexity**
>
> We appreciate your comments on the description of complexity. We will revise it accordingly to avoid any confusion.
>
> As discussed in Section 3, the generalized projection step has a typical computational cost of $\tau_{\text{G-proj}} = O(n^3)$, and we agree that this should have been explicitly stated in Table 1. We will revise the table caption to include this remark.
>
> While ONS’s per-round complexity is indeed higher than that of OGD, it is important to note that existing logarithmic regret methods by Besbes et al. (2021, 2025) and Gollapudi et al. (2021) incur even greater per-round costs that depend on both $n$ and $T$—e.g., $O(n^5T^3)$, as noted in the caption of Table 1. In contrast, our approach eliminates any dependence on $T$ and typically keeps the per-round cost at $O(\tau_{\text{solve}} + n^3)$ when $\tau_{\text{G-proj}} = O(n^3)$, where $\tau_{\text{solve}}$ denotes the cost of solving linear optimization over $X_t$, which is also required by OGD. We thus believe that this complexity should be regarded as a strength of our approach rather than a weakness. In fact, Reviewer 1Hto noted that “This is a significant advancement, as previous logarithmic-regret methods were highly inefficient for large time steps.”
>
> ## **W3. Numerical experiments**
> We appreciate the reviewer's feedback regarding numerical experiments. In light of your and other reviewers’ comments, we have conducted experiments to complement our theoretical results.
>
> ### **Setup**
> We use synthetic datasets inspired by the hard instance used in our lower bound analysis (Section 5). Let $c^\ast$ be a random vector with $\\| c^\ast \\| = 1$. The learner’s prediction set $\Theta$ is the $n$-dimensional Euclidean unit ball. For each $t$, we sample an endpoint $v$ uniformly at random from the unit sphere in $\mathbb{R}^n$ and set $X_t$ to the line segment $[-v, +v]$. We generate $T = 10,000$ instances for $n = 2$, $20$, and $200$.
>
> ### **Methods**
> We compare the following three methods:
> - ONS: our online-Newton-step-based method,
> - OGD: the online gradient descent method based on Bärmann et al. (2020),
> - CP: a cutting-plane-style method inspired by Gollapudi et al. (2021).
>
> For CP, we must compute the centroid of $\bigcap_{s=1}^t \\{c \in \Theta : \langle c - \hat c_s, x_s - \hat{x}_s \rangle \ge 0\\}$ at each round $t$. Since exact centroid computation is #P-hard, and even polynomial-time approximations like hit-and-run sampling are often impractical, we adopt a randomized heuristic: we pre-sample $n \times 10^4$ candidate points from the unit ball $\Theta$, eliminate those violating accumulated cuts, and approximate the centroid as the average of the remaining points. Though not theoretically grounded, this method serves as a tractable approximation of the original CP approach.
>
> ### **Results**
> Below are tables showing the cumulative regret and runtime over $T$ rounds.
>
> #### **Cumulative regret**
>
> | Method         | $n = 2$   | $n = 20$   | $n = 200$  |
> |----------------|---------|----------|----------|
> | ONS            | $4.93$    | $31.69$    | $46.85$    |
> | OGD            | $7.33$    | $33.20$    | $69.43$    |
> | CP             | $3.81$ | $564.49$   | $481.45$   |
>
> #### **Cumulative runtime (s)**
>
> | Method         | $n = 2$   | $n = 20$   | $n = 200$  |
> |----------------|---------|----------|----------|
> | ONS            | $0.137$   | $1.53$     | $17.26$    |
> | OGD            | $0.046$| $0.076$| $0.134$|
> | CP             | $1.64$    | $2.01$     | $41.31$    |
>
> When the dimension is small ($n = 2$), CP achieves the lowest cumulative regret. However, its cumulative regret deteriorates significantly for $n = 20$ and $n = 200$. This degradation stems from the limited number of pre-sampled candidate points: in principle, about $T^n$ samples would be required for an accurate approximation, which is infeasible even for moderate $n$. Moreover, although the above randomized centroid approximation substantially reduces computation by averaging over the surviving candidates, it remains less scalable than OGD or ONS. These results highlight practical limitations of CP in moderate to high-dimensional settings.
>
> In contrast, ONS consistently achieves low regret values across all dimensions while remaining computationally feasible. It outperforms OGD in terms of the regret and scales reasonably well with increasing $n$. Note that our ONS implementation uses a straightforward projection routine that repeatedly solves similar linear systems—an overhead that could be reduced by more sophisticated implementation techniques. Further speedups could also be achieved via quasi-Newton-type updates or sketching-based techniques, as discussed in Sections 3 and 4.
>
> Taken together, these findings affirm that ONS provides a strong and scalable alternative to existing methods in online inverse linear optimization, especially when CP is computationally infeasible and OGD's regret performance is unsatisfactory.
>
> ## **Q. Relation to online learning with hints**
> Thank you for the interesting question. We considered your suggestion in light of the literature on *online learning with hints*, such as Dekel et al. (NeurIPS 2017, "Online Learning with a Hint") and Bhaskara et al. (ICML 2020, "Online Learning with Imperfect Hints"). Below, we clarify the similarities and differences.
>
> The "online learning with hints" framework pertains to standard online linear optimization (OLO), where the learner selects an action $x_t$ and then observes a cost vector $c_t$. Before making a decision, the learner receives a hint vector $h_t$, which is expected to correlate with $c_t$. The goal is to minimize the regret $\sum_{t=1}^T \langle c_t, x_t - u \rangle$ with respect to the best fixed action $u$ in hindsight.
>
> In contrast, our setting arises from inverse optimization. The learner predicts an objective vector $\hat{c}\_t$, which induces an action $\hat{x}\_t \in \mathrm{argmax}\_{x \in X_t} \langle \hat{c}_t, x \rangle$, and then observes the agent’s actual choice $x\_t \in X\_t$. The regret is defined as $\sum\_{t=1}^T \langle c^{\ast}, x_t - \hat{x}\_t \rangle$, where $c^{\ast}$ is the agent's true objective vector. Thus, the learner selects $\hat{c}_t$, and the feedback is observed *after* decision-making, in the form of the agent’s action $x_t$.
>
> Although the two frameworks might appear similar, they differ in fundamental ways. In the hint setting, the learner receives explicit side information *before* making a decision. In our setting, the feedback $x_t$ arrives *after* the learner’s prediction, and it only indirectly reflects the unknown objective vector $c^\ast$.
>
> That said, one could envision a hybrid setup where the learner receives both the agent's feedback and external side information about $c^{\ast}$. Developing algorithms that leverage such additional information—perhaps to remove the $\log T$ factor in the regret bound—would be an exciting direction for future research. We thank the reviewer for raising this intriguing connection and will include it in the revised version.
>
> We hope the above clarifications help resolve your concerns. Please do not hesitate to let us know if any questions remain or if further clarification would be helpful.

---

> > ### Comment · Reviewer_xL4W · 2025-08-05
> >
> > I would like to thank the authors for their responses. I do not have any further questions for the authors, and I will wait for the discussion with other reviewers.

---

### Official Review · Reviewer_1Hto · 2025-07-02

**Clarity:** 2
**Significance:** 3
**Originality:** 3
**Rating:** 4
**Confidence:** 4

**Summary:**

This paper aims to solve the problem of online inverse linear optimization, where a learner tries to deduce a hidden linear objective function of an agent by observing the agent's chosen optimal actions over time.
The quality of the learner's predictions is measured by regret, which is the cumulative gap between optimal objective values and those achieved by following the learner's predictions.
In the literature, Besbes et al. 2021 and 2025 propose an algorithm that achieves a logarithmic regret bound, while it is not computationally efficient since the number of constraints grows as fast as a polynomial of time length $T$. The authors propose an efficient algorithm that achieves logarithmic regret $O(n \ln(T)$ and has a per-round complexity independent of $T$.

**Questions:**

1. Line 290 - 296, you state that MetaGrad's per-round complexity is $O(\tau_{\text{solve}} + (n^2 + \tau_{G-\text{proj}}) \ln(T)$ and that "If the $\tau_{G-\text{proj}} \ln(T)$ factor is a bottleneck, we can use more efficient universal algorithms [Mhammedi et al., 2019, Yang et al., 2024] to reduce the number of projections from $\Theta(lnT)$ to $1$."  Could you briefly explain how these more efficient universal algorithms (e.g., [Mhammedi et al., 2019, Yang et al., 2024]) reduce the number of projections to 1 in the context of MetaGrad? And what is the factor or concern that prevents you from choosing those more advanced algorithms in the first place?

2. Line 338 - 339, you mention, "Whether a similar lower bound holds when all $X_t$ are full-dimensional remains an open question." Can you briefly speculate on what specific challenges arise in constructing a similar lower bound proof for the full-dimensional case compared to the line segments used in Theorem 5.1?

3. Line 715 - 727, in Section C.2, you discuss the challenge of extending the $O(1)$ regret bound for $n=2$ to $n \geq 3$ due to the area $A(C_t)$ becoming arbitrarily small while $\theta(c^\star, \hat{c})$ does not. Could you elaborate more on the most promising direction you believe to close the gap for $n \geq 3$?

**Ethical Concerns:**

["NO or VERY MINOR ethics concerns only"]

**Final Justification:**

The author has resolved my questions. I have also reviewed the discussions of the author with other reviewers, and I believe this paper is acceptable so I will keep my score and support for this paper.

**Limitations:**

yes

**Paper Formatting Concerns:**

No obvious formatting concerns were found in the paper.

**Quality:**

3

**Strengths And Weaknesses:**

Strengths:

1. The paper presents the first logarithmic-regret method whose per-round computational complexity is independent of the time horizon $T$. This is a significant advancement, as previous logarithmic-regret methods were highly inefficient for large time steps.
Also, the proposed method achieves the best-known logarithmic regret bound of $O(n\ln (T))$

2. The method is described as "strikingly simple" as it applies the Online Newton Step (ONS) to appropriate exp-concave loss functions. This simplicity is considered a strength, making it more accessible and easier to implement.

3. Corollary 4.2 provides an online-to-batch conversion for offline settings. The comparison with [Bärmann et al., 2020] on offline guarantees notes that their prior work is $O(1/T)$ even when $\Delta=0$, whereas this paper offers a faster $O(\ln(T)/T)$ rate, highlighting an improvement.

Weakness:

1. The primary limitation explicitly stated by the authors is that their work is restricted to the case where the agent's underlying optimization problem is linear. Extending this to nonlinear settings is identified as an important direction for future work.

2. While the lower bound of $\Omega(n)$ and upper bound of $O(n\ln(T))$ are established, there remains an $O(\ln(T))$ gap between them that the authors identify as an important open problem.

---

> ### Author Rebuttal · Authors · 2025-07-25
>
> We sincerely thank the reviewer for the insightful comments and questions. Below, we address each point raised.
>
> ## **Q1. Reducing the number of projections in MetaGrad**
> The reason we used the standard version of MetaGrad in the complexity discussion is that it is both intuitive and widely known. More efficient implementations in Mhammedi et al. (2019, Section 4) can indeed reduce the computation cost of projections from that of $O(\log T)$ to 1 per round without any sacrifice, which we explain below.
>
> In MetaGrad, each of the $O(\log T)$ $\eta$-experts performs a generalized projection of the following form, as in Algorithm 1 in Appendix B:
>
> $$
> w_{t+1} \leftarrow \arg\min_{w \in \mathcal{W}} \left\\| w_t - \frac{1}{\gamma} A_t^{-1} \nabla q_t(w_t) - w \right\\|_{A_t}^2,
> $$
>
> where $A_t = \varepsilon I_n + S_t$ and $S_t = \sum_{s=1}^t \nabla q_s(w_s) \nabla q_s(w_s)^\top$.
> Let $\mathcal{W}$ be a Euclidean ball for simplicity. Then, the projection can be computed via singular value decomposition (SVD) of $A_t$. Naively doing this for each expert would require $O(\log T)$ SVD computations. However, as observed in Mhammedi et al. (2019, Section 4.1), $S_t$ is the sum of outer products of the same vectors across all $\eta$-experts, and the only difference among the experts lies in the scalar weighting applied to $S_t$. This enables us to compute the SVD of $S_t$ once and reuse it across all experts, thereby reducing the projection cost to that of a single projection. Even when $\mathcal{W}$ is not a Euclidean ball, the approach can be extended by reducing the projection to the Euclidean case based on a technique developed by Cutkosky and Orabona (2018, "Black-box reductions for parameter-free online learning in Banach spaces"), as discussed in Mhammedi et al. (2019, Section 4.2).
>
> ## **Q2. On extending the lower bound to full-dimensional $X_t$**
> Thank you for highlighting this subtle and important point. Our current lower bound construction relies on using line segment domains $X_t$ that are axis-aligned. Intuitively, this structure ensures that the observed feedback $x_t \in \mathrm{argmax}_{x \in X_t} \langle c^\ast, x \rangle$ reveals only a single coordinate of the agent’s true objective vector $c^\ast$. For example, if $X_t$ is aligned with the $t$-th coordinate axis, then $x_t$ provides information only about $c^*(t)$ and nothing about the other coordinates. This readily implies that $\Omega(n)$ observations are required to infer the full vector $c^\ast$, which forms the basis of our lower bound.
>
> In contrast, if $X_t$ is full-dimensional—say, $X_t = [-1, 1]^2$ for $n = 2$—the maximizer $x_t$ lies at one of the box’s corners. Observing the sign pattern of $x_t$ in such cases allows the learner to infer the sign of every coordinate of $c^*$, thereby revealing more information per round than in the line segment case.
>
> Establishing lower bounds under full-dimensional $X_t$ thus involves additional challenges in limiting per-round information gain. For the purposes of this work, using line segment domains is sufficient to derive the $\Omega(n)$ lower bound, while keeping the proof technically simple and transparent.
>
> ## **Q3. On the discussion in Appendix C**
> We are glad that the reviewer found the discussion in Appendix C of interest. This part of our analysis is inspired by cutting-plane-type ideas (Besbes et al., 2021, 2025; Gollapudi et al., 2021), where the learner iteratively eliminates inconsistent regions of the objective vector space. However, as noted, this approach faces technical barriers—such as the gap between angular and area—that make it challenging to extend the analysis to general $n$.
>
> In contrast, our main approach builds on online Newton step (ONS) with tailored exp-concave surrogate losses. While this incurs a $\log T$ factor, the online convex optimization (OCO) literature provides data-dependent refinements for exp-concave losses. For example, Orabona et al. (AISTATS 2012, "Beyond Logarithmic Bounds in Online Learning") introduce the so-called small-loss-dependent analysis, which tightens the regret bound based on the comparator's cumulative loss. Adapting such advanced OCO techniques to online inverse optimization could be a promising future direction. More broadly, we believe that the cutting-plane and ONS-based viewpoints highlight complementary aspects of the problem, and integrating insights from both may ultimately help close the $\log T$ gap.
>
> We hope the above responses effectively address your questions. Please do not hesitate to let us know if any questions remain or if further clarification would be helpful.

---

> > ### Comment · Reviewer_1Hto · 2025-08-04
> >
> > I particularly want to thank to author for their detailed feedback, and they have resolved my questions. I also go through other reviewers' feedback and rebuttals. I would like to keep my support for this paper and maintain my score.

---

### Official Review · Reviewer_kNBn · 2025-07-02

**Clarity:** 2
**Significance:** 3
**Originality:** 3
**Rating:** 5
**Confidence:** 4

**Summary:**

This papers studies the problem online inverse linear optimization, proposing an algorithm with an improved regret bound of $O(n \ln T)$  with a simple and relatively efficient algorithm. The main idea is the use of the Online Newton Step with a surrogate function proposed by the authors of the algorithm MetaGrad. Furthermore, the authors show that when the learner observes a suboptimal action instead of an optimal one as in the case of the original problem, using MetaGrad with $\Theta(\log T)$ experts leads to a similar regret bound with an extra penalty of $O(\sqrt{\Delta_T n \ln T})$, where $\Delta_T$ is the culmulative gap of the optimal payoffs and the payoffs of the observed actions. Finally, the authors also show a $\Theta(n)$ lower bound on the regret of any (possibly randomized) algorithm for the problem, and show how one can get $O(1)$ regret for $n = 2$.

**Questions:**

Here are a few parts where notation is heavy and maybe a bit confusing:
-  In Prop 2.5, do we need $w_i \in \mathcal{W}$? Or is it ok for them to be arbitrary vectors? Also, having $w_t$ as arbitrary vectors while $w_t^{\eta}$ are iterates was quite confusing. It makes sense for MetaGrad, but at this point it was quite confusing;
- For Theorem 4.1 we need the subgradients used by MetaGrad to be the specific gradients given by Prop 2.4, right? I don't think this is specified anywhere
Other minor comments:
- lines 49-50: "(...) about $c^*$ defining the regret." not clear what you meant here
- line 138: "Alternately" here is to mean that first the player places the prediction and only afterwards receives feedback, if I understood it correctly, but I don't think this is the correct way to say that.
- Assumption 2.2: "$\ell_2$-diameter" is slightly ambiguous since we don't know if you mean squared norm or not. If you have the space, I would add the mathematical formula to clear any possible confusions;

**Ethical Concerns:**

["NO or VERY MINOR ethics concerns only"]

**Final Justification:**

I went throught the other reviews, and I still believe this is a strong submission. I don't agree with the most negative review, and the other slightly negative reviewer had some good points that I felt were properly addressed in the rebuttal. I'll engage in the discussion with reviewers and AC to try to reach consensus.

**Limitations:**

The author's do a good job on placing their contributions in the literature and discussing limitations.

**Paper Formatting Concerns:**

No formatting concerns

**Quality:**

3

**Strengths And Weaknesses:**

The algorithm proposed to the problem is surprisingly simple and yields strong results for the problem. It is quite interesting to see such a simple algorithm given that the previous algorithms for the same (or similar problem) use more intricate algorithms based on cutting plane methods. Ultimately the regret bounds follow from results from MetaGrad (as the authors note, even Thm 3.1 is MetaGrad with only one expert), but this should not be seen as a negative point for this paper, since this application was not obvious at all in hindsight and the authors do shave off some constants in their analysis. The lower bound is also simple, so I feel the results overall are quite elegant.

One problem (although relatively minor) the paper has is with notation. It is quite heavy at some moments, and it can get quite confusing at some points. I know the MetaGrad paper and know that a lot of this notation is from the original MetaGrad paper, but this is not a good excuse to use the same notation in the main paper.

Another minor problem with presentation is that it was not clear to me when first reading the paper whether the constants the big-Oh notation was hiding in sec 1 could be hiding a dependency on $n$ (such as the B or D constants later on). Appendix A (which is quite well-written) clarifies this a lot, but it would be nice to say, for example, that the rates in the table in Sec 1 are for all sets being constrained to the unit ball for the sake of uniformity.

---

> ### Author Rebuttal · Authors · 2025-07-25
>
> We sincerely thank the reviewer for the careful and constructive feedback, as well as for your strong support of our work. We are grateful for the detailed reading and thoughtful suggestions rooted in deep expertise. Below, we respond to the questions:
>
> 1. Regarding the domain of $w_t$ in Proposition 2.5, the vector (as well as $w_t^\eta$) is assumed to belong to $\mathcal{W}$. In the main text, $w_t \in \mathcal{W}$ is stated in eq. (3).
>
> 2. In Theorem 4.1, subgradient $g_t$ corresponds to $\hat{x}_t - x_t$, consistent with Proposition 2.4. We will clarify this in the revised version.
>
> We also appreciate the minor comments and will incorporate the suggested improvements in the revised version.
>
> Regarding the remark on lines 49–50: we intended to convey that online inverse linear optimization can, in principle, benefit from richer feedback than standard online linear optimization (OLO). In OLO, the cost vectors provide little information about the best action in hindsight, which is the comparator in the regret, restricting the attainable regret rate to $\sqrt{T}$. In contrast, in online inverse linear optimization, the observed feedback $x_t \in X_t$ is optimal for the agent’s true objective vector $c^\ast$, which defines the regret in this setting as $\sum_{t=1}^T \langle c^\ast, x_t - \hat x_t \rangle$. Intuitively, this richer information about $c^\ast$ enables us to achieve logarithmic regret bounds, surpassing the $\sqrt{T}$ barrier of general OLO.

---

> > ### Comment · Reviewer_kNBn · 2025-08-04
> >
> > I would like to thank the authors for their rebuttal to my review and to the other reviewers as well, it helps a lot to see the discussion in other reviews.
> >
> > About the (minor) points I raised, I think the longer discussion the authors provided about the comments on lines 49-50 does help a lot. In the original write-up I had a hard time interpreting it just by the way it is framed. If space allows, I would encourage incorporating this lengthier version to the paper, since it is a nice discussion when introducing the problem.
> >
> > I went throught the other reviews, and I still believe this is a strong submission. I don't agree with the most negative review, and the other slightly negative reviewer had some good points that I felt were properly addressed in the rebuttal. I'll engage in the discussion with reviewers and AC to try to reach consensus.

---

### Official Review · Reviewer_5DsP · 2025-07-02

**Clarity:** 3
**Significance:** 2
**Originality:** 2
**Rating:** 3
**Confidence:** 2

**Summary:**

This paper addresses the problem of online inverse linear optimization, where the learner is given a sequence of feasible sets $X_t \subseteq \mathbb{R}^n$ and corresponding optimal decisions $x_t$, and seeks to recover the unknown linear cost vector $c^*$. The authors propose a learning algorithm based on the Online Newton Step (ONS) method applied to a specially constructed loss function. Assuming this loss is exp-concave, the algorithm achieves a regret bound of $O(n \log T)$, with per-round computational complexity independent of the time horizon $T$. The algorithm is also extended to handle suboptimal actions using a MetaGrad-style ensemble. A lower bound of $\Omega(n)$ is provided, establishing the optimality of the regret's dimensional dependence up to logarithmic factors.

**Questions:**

1. Can you provide a formal proof or structural conditions under which the proposed loss function is exp-concave? Without this, the regret bound in Theorem 3.1 is not well grounded.
2. Have you considered implementing your algorithm and comparing its runtime and regret behavior to prior methods (e.g., Besbes et al., which accumulate constraints over time)? Even synthetic experiments could strengthen the paper.
3. The regret bounds include constants tied to parameters like $B, D, K$. Can you give concrete estimates or examples that help interpret these constants in typical inverse optimization settings?
4. This work is clearly focused on linear objectives. Section 6 mentions the potential for kernelization. Can your self-bounding analysis extend to non-linear objectives or convex settings? Is exp-concavity still plausible in those cases?
5. The $\log T$ gap between upper and lower bounds remains open. Is this tight, or can the upper bound be improved with further algorithmic refinements?

**Ethical Concerns:**

["NO or VERY MINOR ethics concerns only"]

**Final Justification:**

I thank the authors for the detailed responses that addressed my concerns/problems. The exp-concavity of the surrogate loss function is clearly proved and the simplified analysis technique can be seen as a strength. Upon reading through other reviewers' comments, I decided to increase my score while decrease the confidence.

**Limitations:**

While the paper discusses the restriction to linear objectives and the unresolved $\log T$ gap, it does not sufficiently highlight the reliance on the unproven exp-concavity assumption, which is critical to the validity of the main result. This should be addressed explicitly.

**Quality:**

2

**Strengths And Weaknesses:**

### Strengths

1. The paper studies a well-motivated and practically relevant problem in online learning and inverse optimization.
2. The paper includes a matching lower bound, which strengthens the theoretical framing.

### Weaknesses

1. The central theoretical guarantee relies entirely on the assumption that the proposed loss function is exp-concave. However, the paper does not prove this property or provide sufficient conditions under which it holds. Since exp-concavity is a strong assumption that does not hold generically, the regret bound is not fully justified.
2. Once exp-concavity is assumed, the regret analysis is a standard application of the ONS algorithm. The work does not introduce new techniques or insights beyond this reuse.
3. The paper includes no experiments to demonstrate the method’s practical efficiency or its advantage over existing baselines such as constraint-accumulation methods. This significantly limits its potential impact.
4. The regret bound hides constants involving quantities such as $B, D, K$, which are not analyzed or estimated. This limits the practical interpretability of the results.
5. Although the paper focuses on linear objectives, the framing and claims suggest broader applicability. In reality, the approach is tailored to a narrow setting, and the key assumptions do not generalize easily.

---

> ### Author Rebuttal · Authors · 2025-07-24
>
> We sincerely thank the reviewer for their time and effort. Feedback that articulates specific points, like yours, is especially helpful in refining the manuscript. Below, we address the questions and concerns.
>
> ## **Q1. Proof of exp-concavity**
>
> The loss function used in Theorem 3.1 is exp-concave under Assumption 2.2. The proof is given in Appendix B.2 (as part of the general proof of Proposition 2.5), and we reproduce it below for clarity.
>
> Specifically, for the $t$-th feedback $x_t \in X_t$, prediction $\hat c_t \in \Theta$, and $\hat x_t \in \mathrm{argmax}_{x \in X_t} \langle \hat c_t, x \rangle$, the loss function is defined as
> $$
> \ell_t^\eta(c) = -\eta \langle \hat{c}_t - c, \hat{x}_t - x_t \rangle + \eta^2 \langle \hat{c}_t - c, \hat{x}_t - x_t \rangle^2 \quad \forall c \in \Theta,
> $$
> where $\eta = \frac{1}{5B}$. Its gradient and Hessian are
> $$
> \nabla \ell_t^\eta(c) = \eta\left(1 - 2\eta \langle \hat{c}_t - c, \hat{x}_t - x_t \rangle \right)(\hat{x}_t - x_t), \quad
> \nabla^2 \ell_t^\eta(c) = 2\eta^2 (\hat{x}_t - x_t)(\hat{x}_t - x_t)^\top.
> $$
>
> Therefore, we have
> $$
> \nabla \ell_t^\eta(c) \nabla \ell_t^\eta(c)^\top = \eta^2\left(1 - 2\eta \langle \hat{c}_t - c, \hat{x}_t - x_t \rangle \right)^2(\hat{x}_t - x_t)(\hat{x}_t - x_t)^\top
> \preceq \eta^2\left(1 + 2\eta B\right)^2(\hat{x}_t - x_t)(\hat{x}_t - x_t)^\top
> = \frac{\left(1 + 2\eta B\right)^2}{2}\nabla^2\ell_t^\eta(c),
> $$
> where the inequality uses Assumption 2.2, which ensures $|\langle \hat{c}_t - c, \hat{x}_t - x_t \rangle| \le B$. Thus, $\ell_t^\eta$ satisfies $\nabla^2\ell_t^\eta(c) \succeq \alpha \nabla \ell_t^\eta(c) \nabla \ell_t^\eta(c)^\top$ for $\alpha = \frac{2}{\left(1 + 2\eta B\right)^2} = \frac{50}{49} > 1$.
>
> This condition is equivalent to $\alpha$-exp-concavity for the twice-differentiable function $\ell_t^\eta$ (e.g., Hazan [26, Lemma 4.2]). Specifically, the Hessian of the function $\phi\colon c \mapsto \exp(-\alpha\ell^\eta_t(c))$ satisfies
> $$
> \nabla^2 \phi(c) = \alpha \exp(-\alpha \ell^\eta_t(c)) \left(\alpha\nabla \ell^\eta_t(c) \nabla \ell^\eta_t(c)^\top - \nabla^2 \ell^\eta_t(c)\right) \preceq 0,
> $$
> which implies that $\phi$ is concave. Therefore, $\ell_t^\eta$ is $\alpha$-exp-concave for $\alpha > 1$.
>
> **Novelty given the exp-concavity.** While our methods build on ONS and MetaGrad, the way we leverage these tools in the context of online inverse linear optimization is novel and nontrivial. A key technical contribution is the identification of the exp-concave surrogate loss that enables the simple yet effective use of ONS in this setting. Notably, commonly used losses in inverse optimization—such as the suboptimality loss—are convex but not exp-concave, and thus do not support logarithmic regret bounds via ONS. Regarding this point, Reviewer 1Hto noted that “... it applies the Online Newton Step (ONS) to appropriate exp-concave loss functions. This simplicity is considered a strength, ...” More importantly, this viewpoint leads to our second main result: robustness to suboptimal feedback via MetaGrad. Reviewer kNBn remarked this application was “not obvious at all in hindsight,” highlighting the conceptual and technical novelty of our approach.
>
> We hope this clarification resolves the reviewer’s concerns regarding the exp-concavity and the technical novelty of our contributions.
>
> ## **Q2. Experiments**
> Thank you for raising this point. Encouraged by your question, we conducted numerical experiments to evaluate our approach empirically. In summary, we found that our ONS-based method consistently outperforms OGD (Bärmann et al., 2020) in terms of the regret, while remaining significantly more efficient than cutting-plane-style methods, akin to those proposed by Besbes et al. (2021, 2025) and Gollapudi et al. (2021). These findings suggest that our ONS-based method is a highly effective approach, particularly when the problem dimension is too high for cutting-plane methods to be feasible and when OGD’s regret performance is unsatisfactory. Due to space limitations, we present the details of the experimental setup and results in our response to Reviewer xL4W.
>
> ## **Q3. Regarding constants depending on $D$, $K$, and $B$**
>
> Our regret upper bound of $O\left( Bn \log \frac{DKT}{Bn} \right)$ depends only mildly on the parameters $B$, $D$, and $K$. The constant factor is interpretable and can be estimated based on the agent’s forward problem specifications. Let us first clarify their definitions:
> - $D$ is the $\ell_2$-diameter of $\Theta$, the learner’s set of predictions.
> - $K$ is the maximum $\ell_2$-diameter of $X_t$, the agent’s feasible action set.
> - $B$ is an upper bound on the inner product $\langle \hat{c} - c, \hat{x} - x \rangle$ for any $c, \hat{c} \in \Theta$ and $x, \hat{x} \in X_t$.
>
> In many cases, the learner is interested only in predicting the direction of the agent’s objective vector $c^\ast$, since the observed feedback $x_t \in \mathrm{argmax}_{x \in X_t} \langle c^\ast, x \rangle$ reveals no information about the scale of $c^\ast$. Therefore, we can usually assume that $\Theta$ is the unit Euclidean ball, yielding $D = 2$. If $c^\ast$ is known to be non-negative, we may alternatively assume that $\Theta$ is, for example, the probability simplex, as in Bärmann et al. (2017).
>
> The value of $K$ depends on the agent’s forward problem. For instance, in the customer preference estimation problem studied by Bärmann et al. (2017, Section 4.1), the action set is defined by a knapsack constraint: $p^\top x \le b$ and $x \ge 0$, for some price vector $p \in \mathbb{R}^n_{>0}$ and budget $b > 0$. If $l > 0$ is a lower bound on the price vector entries, then the $\ell_2$-diameter is at most $K = O(b\sqrt{n}/l)$. In the literature (e.g., Gollapudi et al., 2021; Besbes et al., 2021, 2025), this is often simplified to $K = O(1)$ by normalizing the action sets.
>
> Finally, the constant $B$ is upper bounded by $DK$ by the Cauchy–Schwarz inequality. We introduce $B$ to express the regret bounds with sharper dependence on the problem geometry. In practice, $B$ can also be interpreted as prior knowledge about the agent’s maximum attainable objective value (up to a constant factor).
>
> Returning to the $O\left( Bn \log \frac{DKT}{Bn} \right)$ regret upper bound, given a specification of the agent’s forward problem as discussed above, the constants in the bound can be computed straightforwardly. Roughly speaking, the logarithmic term can be estimated as at most $\log T$ since we typically have $DK \approx B$. Thus, the dominant contribution from the constant factor is $Bn$, which can be interpreted as the agent’s maximum attainable objective value multiplied by the dimensionality. Note that this is the optimal dependence on $Bn$ since there is a regret lower bound of $\Omega(Bn)$, as shown in Theorem 5.1.
>
> Appendix A provides a more detailed discussion on how these parameters influence the regret bounds, including comparisons with prior work.
>
> ## **Q4. Extension to non-linear objectives**
> Thank you for the question. While our paper focuses on linear objectives, the techniques extend to a broad class of forward problems.
>
> If the agent's objective function is nonlinear—for example, quadratic of the form $x^\top Q x + q^\top x$—we can still apply our framework by defining a suitable feature map. Specifically, we may define
>
> $$\phi(x) = (x_1^2, \ldots, x_n^2, x_1 x_2, \ldots, x_{n-1}x_n, x_1, \ldots, x_n) \in \mathbb{R}^{\binom{n}{2} + 2n}$$
>
> so that the agent’s objective becomes linear in the feature space: $\langle c, \phi(x) \rangle$ with $c \in \mathbb{R}^{\binom{n}{2} + 2n}$ encoding $Q$ and $q$. As long as the objective is expressed in this linear form in $c$, our surrogate loss, $\ell^\eta_t(c)$, remains exp-concave, and our approach can be applied; we have discussed a similar extension to the contextual setting of Besbes et al. (2021, 2025) in 562–578 in Appendix A. In this sense, our approach is compatible with a wide class of non-linear objectives. The main limitation of this extension is the increased dimensionality. Designing scalable algorithms that mitigate this issue is an important direction for future work.
>
> Finally, note that although the linear setting may appear restrictive, it captures many practical applications and provides a theoretical foundation for understanding the regret guarantees achievable in inverse optimization. This is why a foundational body of prior work—including Bärmann et al. (2017, 2020), Gollapudi et al. (2021), and Besbes et al. (2021, 2025)—has also focused on linear objectives.
>
> ## **Q5. Regarding the $\log T$-gap**
>
> Whether the $\log T$ factor is avoidable in online inverse optimization is a fundamental open question, as also noted in prior work (e.g., Gollapudi et al., 2021, Section 4.1). Although no existing work has formally resolved this question, we believe that further progress could lead to removing the $\log T$ factor from the regret upper bound. Our work contributes to this broader effort by offering a novel perspective: whereas previous approaches achieving logarithmic regret bounds (Gollapudi et al., 2021; Besbes et al., 2021, 2025) are based on cutting-plane methods, our approach is the first to leverage ONS in this context. We hope that this alternative methodology sheds new light on the structure of the problem and offers a fresh angle from which to approach this important open direction.
>
> We hope that the above responses effectively address your concerns. Please do not hesitate to let us know if any questions remain or if further clarification would be helpful.

---

> > ### Comment · Reviewer_5DsP · 2025-08-05
> >
> > I thank the authors for the detailed responses that addressed my concerns/questions. The exp-concavity of the surrogate loss function is clearly proved and the simplified analysis technique can be seen as a strength. Upon reading through other reviewers' comments, I decided to increase my score while decrease the confidence.

---

### Decision · Program_Chairs · 2025-09-17

**Decision:**

Accept (poster)

**Comment:**

This paper studies the online inverse linear optimization problem, and shows that ONS and MetaGrad with suitable surrogate loss functions can achieve the logarithmic regret bounds for this problem. The authors also establish an $\Omega(n)$ lower bound to demonstrate the near optimality of their algorithms, where $n$ is the dimensionality.

Note that previous studies have already proposed an algorithm to achieve the logarithmic regret bound for this problem. However, this algorithm is a cutting-plane-style method, which is more complex and requires much higher time complexity. Therefore, the main strengths of this paper are the efficient logarithmic-regret algorithms (i.e., ONS and MetaGrad with suitable surrogate loss functions) and the $\Omega(n)$ lower bound.

Nonetheless, there are also some weaknesses (or concerns initially raised by some reviewers). First, the applications of ONS and MetaGrad are very standard, as MetaGrad was originally proposed with surrogate loss functions [55, 56], which implies that the technical novelty of this paper may be limited. Second, there still exists a time-dependent gap between the $\Omega(n)$ lower bound and the $O(n\log T)$ upper bound. Third, the authors do not provide experiments to verify the advantage of the proposed algorithms.

Due to these strengths and weaknesses, this paper initially got scores of 2, 4, 4, 3, and 5 from five reviewers. But after the rebuttal, most concerns of the reviewers have been addressed: i) they reach a consensus that the simplicity of applying ONS and MetaGrad should be regarded as a strength, rather than a weakness; ii) the authors provide some experiments to demonstrate the advantage of the proposed algorithms. Accordingly, their scores became 3, 4, 5, 4, and 5.

I also like the simplicity (and efficiency) of applying ONS and MetaGrad to the online inverse linear optimization problem. Thus, I recommend accepting this paper, and hope that the authors could also revise this paper according to the reviewers' comments, e.g., the concerns about the experiments.